# Activation of Prp28 ATPase by phosphorylated Npl3 at a critical step of spliceosome remodeling

Fu-Lung Yeh[1], Shang-Lin Chang [1], Golam Rizvee Ahmed [1], Hsin-I Liu[1], Luh Tung[1], Chung-Shu Yeh [1], Leah Stands Lanier[2], Corina Maeder [3], Che-Min Lin [1], Shu-Chun Tsai[1], Wan-Yi Hsiao[1,4], Wei-Hau Chang [5] & Tien-Hsien Chang [1✉]

Splicing, a key step in the eukaryotic gene-expression pathway, converts precursor messenger RNA (pre-mRNA) into mRNA by excising introns and ligating exons. This task is accomplished by the spliceosome, a macromolecular machine that must undergo sequential conformational changes to establish its active site. Each of these major changes requires a dedicated DExD/H-box ATPase, but how these enzymes are activated remain obscure. Here we show that Prp28, a yeast DEAD-box ATPase, transiently interacts with the conserved 5′ splice-site (5′SS) GU dinucleotide and makes splicing-dependent contacts with the U1 snRNP protein U1C, and U4/U6.U5 tri-snRNP proteins, Prp8, Brr2, and Snu114. We further show that Prp28's ATPase activity is potentiated by the phosphorylated Npl3, but not the unphosphorylated Npl3, thus suggesting a strategy for regulating DExD/H-box ATPases. We propose that Npl3 is a functional counterpart of the metazoan-specific Prp28 N-terminal region, which can be phosphorylated and serves as an anchor to human spliceosome.

[1] Genomics Research Center, Academia Sinica, Taipei, Taiwan. [2] Department of Biology, Washington and Lee University, Lexington, VA, USA. [3] Department of Chemistry, Trinity University, San Antonio, TX, USA. [4] Institute of Biochemistry and Molecular Biology, National Yang Ming Chiao Tung University, Taipei, Taiwan. [5] Institute of Chemistry, Academia Sinica, Taipei, Taiwan. ✉email: chang108@gate.sinica.edu.tw

Splicing is an essential step in the eukaryotic gene-expression pathway that converts pre-mRNA into mRNA by excising introns and ligating exons. This task, which demands single-nucleotide precision, is accomplished by the spliceosome, a macromolecular machine made of five small nuclear RNAs (snRNAs) and numerous proteins[1–3]. Unique among ribonucleoprotein (RNP) machines, the spliceosome is assembled anew upon each intron and undergoes sequential conformational changes to establish its active site. Each of these major changes[1–3] requires a dedicated DExD/H-box ATPase[4], but how these enzymes are rigorously regulated to trigger specific conformational changes remain obscure.

Spliceosome assembly follows a canonical pathway[1–3] in which the pre-mRNA 5′SS is first recognized by U1 snRNP to form the early (E) complex, which then progresses through complexes A (containing U1 and U2 snRNPs), pre-B (U1, U2, and U4/U6.U5 tri-snRNP), B (U1 snRNP departed), B^act (U4 snRNP departed; 1st chemical step activation), B* (5′SS cleavage and lariat intron formation), C (2nd step activation), C* (3′SS cleavage), and P (product complex with ligated exons and lariat intron). Release of the spliced mRNA creates the intron lariat spliceosome (ILS; containing U2/5/6 and lariat intron), which is dissembled to recycle the snRNPs and degrade the intron. At almost every step of the way, one or two DExD/H-box ATPases are dedicated to driving this pathway forward[4]. For example, Prp28 acts on pre-B

complex to promote a U1/U6 switch at the 5′SS[5,6] such that the U6 snRNA's invariant ACAGAGA box can bind to 5′SS, whereupon Brr2 unwinds the U4/U6 RNA duplex within the B complex to liberate U4 snRNP.

We have previously shown that yeast Prp28 (hereafter Prp28) can be made dispensable upon weakening U1 snRNP/5′SS interaction by specific alterations of U1 snRNP components, including U1C protein and U1 snRNA[6,7], suggesting these components are Prp28's molecular targets. Prp28, however, differs from its human counterpart (hPrp28) by lacking a metazoan-specific N-terminal region[8] and by existing in free form rather than being a part of U5-snRNP[9]. Both purified Prp28 and hPrp28 exhibit negligible ATPase activity[10,11], raising the hypothesis that a cofactor, a spliceosomal environment, or both are required to turn on their activities in the right place at the right time. The present study seeks to investigate how Prp28 comes in contact with the spliceosome and how its ATPase activity can be potentiated within the spliceosome environment.

## Results and discussion

**Prp28 contacts several key spliceosomal proteins during splicing.** To explore the preceding hypothesis, we first examined Prp28's presence in various splicing complexes enriched by low ATP concentrations[12] (Fig. 1a, lanes 11–15 and Supplementary

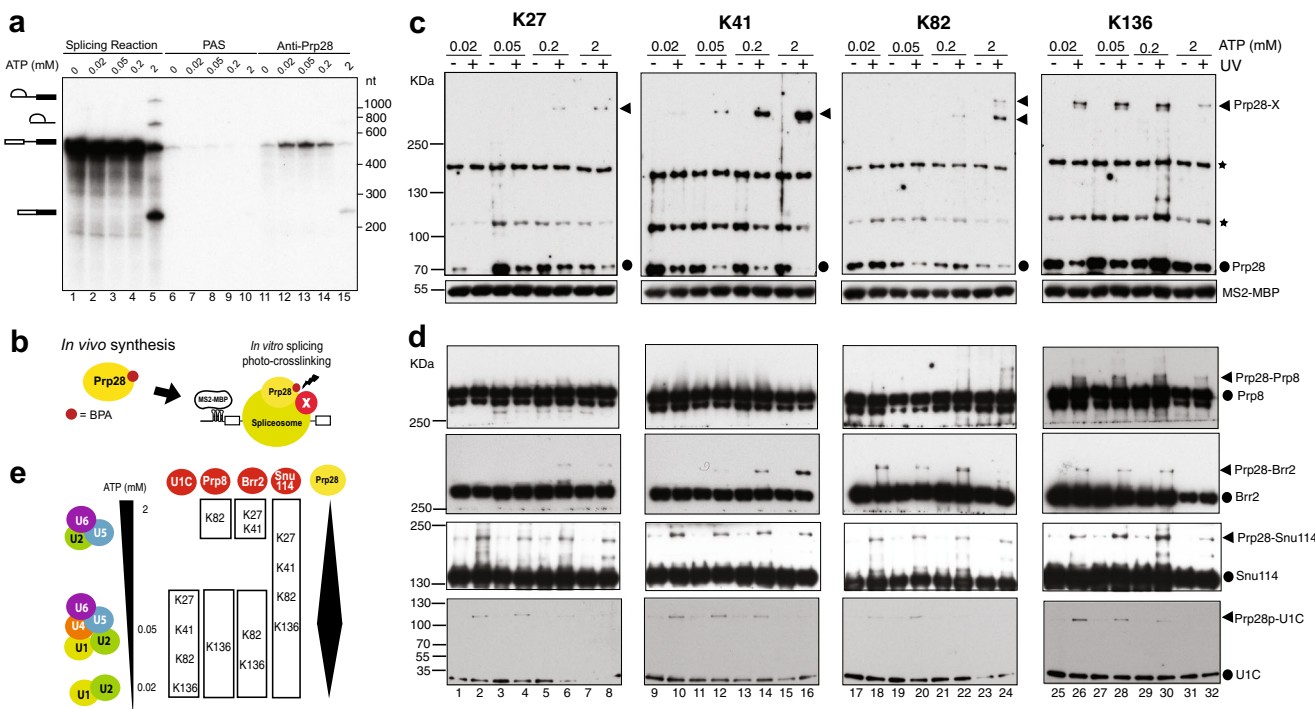

**Fig. 1 Prp28 contacts key proteins at the heart of spliceosome. a** Low ATP traps a transient interaction of Prp28 with the spliceosome. Splicing reactions (lanes 1–5) were done in V5-tagged Prp28 extracts at 0, 0.02, 0.05, 0.2, or 2 mM ATP and a portion was subjected to immunoprecipitation without antibody (PAS; lanes 6–10) or with anti-V5 antibody (lanes 11–15). Relative loadings are 1:10 for splicing reactions alone (lanes 1–5) vs. immunoprecipitated reactions (lanes 6–15). Positions of pre-mRNA, splicing intermediates, and mRNA are indicated to the left. The experiment was repeated three times with similar results. **b** Schematic diagram showing a BPA-marked Prp28 cross-linked to protein X in a spliceosome assembled on MS2 stem-loop-tagged *ACT1* pre-mRNA, which can be pulled down by MS2-maltose-binding protein-(MS2-MBP)-conjugated agarose beads. Thunderbolt, 365-nm UV irradiation. **c** Prp28-BPA cross-linked species (Prp28-X) detected by using anti-Prp28, or using anti-HA and anti-V5 tag antibody for Prp28-tagged experiments. K27, K41, K82, and K136 are the amino-acid residues in Prp28 replaced by BPA. (−) and (+), without or with UV irradiation, respectively. Filled circle, uncrosslinked Prp28. Asterisk, nonspecific background band. Detection of MS2-MBP serves as a loading control. The experiments were repeated three times with similar results. **d** Identification of the X proteins as Prp8, Brr2, Snu114, and U1C by using anti-Prp8, anti-Brr2, anti-Snu114, or anti-V5 (U1C-V5) antibody, respectively. The experiments were repeated three times with similar results. **e** Schematic summary of the cross-linking data. Splicing complexes accumulated at various ATP concentrations are shown to the left. The changing amount of Prp28 associated with the spliceosome is depicted to the right. Source data are provided as a Source Data file.

Fig. 1a, b). The amount of the pre-mRNA co-precipitated by anti-Prp28 antibody peaked at 0.05 mM ATP (Fig. 1a, lane 13). A reciprocal experiment using MS2 loop-tagged pre-mRNA for pulling down splicing complexes yielded the same result (Supplementary Fig. 1a, b). Together, these data suggest that ATP hydrolysis is required for releasing Prp28 from splicing complexes (Fig. 1e and Supplementary Fig. 1c).

To understand Prp28's action within the protein-rich RNP environment of the spliceosome[13], we adapted a *p*-benzoyl-phenylalanine (BPA; a photoactivatable unnatural amino acid) based cross-linking method for detecting potentially transient protein-protein interactions[14,15]. In this approach, BPA is site-specifically incorporated in vivo into Prp28 using an orthogonal pair of aminoacyl tRNA synthetase and suppressor tRNA (UAG). There were several considerations for choosing which amino-acid residues for BPA replacement. We first selected hydrophilic amino-acid residues that are not strictly conserved among DExD/H-box proteins, arguing that they are more likely to situate on Prp28's surface and that their BPA replacements are less likely to significantly impact on Prp28's function. We then used the structural information of Vasa[16], another DExD/H-box protein, to computationally model Prp28 structure and to guide our final selections[17]. Among the 42 UAG (stop codon)-containing *PRP28* alleles tested, 36 supported cell growth in BPA-containing media (Supplementary Data 4). We next prepared active splicing extracts (Supplementary Fig. 1d) from these engineered strains for performing BPA-mediated protein-protein cross-linking (Fig. 1b). Among the 12 extracts that yielded detectable Prp28-cross-linked products, we found that most of those BPA-replaced residues are located on the surface of RecA1 domain or in the N-terminal region of Prp28 that is not resolved in the crystal structure[10] (Supplementary Fig. 2). Data from the Prp28-K27[BPA], -K41[BPA], -K82[BPA], and -K136[BPA] experiments are shown in Fig. 1c. These cross-linked species are splicing-dependent because their appearances depend on the presence of pre-mRNA, intron, functional 5′ SS and branch site, and UV irradiation (Fig. 1c and Supplementary Fig. 3). The addition of RNase A after UV irradiation did not abolish the cross-linking signals, suggesting that Prp28 makes direct contacts with targeted proteins (Supplementary Fig. 3). We then scaled up the Prp28-K136[BPA] reaction for mass-spectrometry analysis (Supplementary Fig. 4), which led to the identification of Prp8, a very large splicing factor in the spliceosome[18] (Supplementary Fig. 4 and Supplementary Data 5). Immunoblotting using anti-Prp8 and anti-Prp28 antibodies confirmed this finding (Supplementary Figs. 1e, f and 4c). On the basis of a combination of cross-linked species' molecular sizes, Prp8's location in published U4/U6.U5 tri-snRNP structures[19,20], and Prp28's known genetic interactions[6,7], we systematically interrogated other cross-linked proteins using a panel of antibodies. This effort identified two additional U5-snRNP proteins, Brr2 and Snu114, as well as U1C (Fig. 1d and Supplementary Fig. 3). There are, however, several other cross-linked species (Supplementary Fig. 1g) that remain to be identified.

To gain insight into Prp28's interactions with these four proteins during spliceosomal assembly, we performed cross-linking experiments by varying ATP concentrations ranging from 0.02 to 2 mM (Fig. 1c–e), which yielded several key observations. First, Prp28 indeed contacts U1C as predicted[6,7], but only at ATP concentrations below 2 mM ATP, consistent with U1 snRNP's departure prior to the occurrence of splicing chemistry at 2 mM ATP[5,6]. The observed Prp28/U1C interaction at 0.02 mM ATP may correspond to Prp28's ATP-independent role in stabilizing early splicing complexes[21]. Second, Prp28 can contact Prp8, Brr2, and Snu114 (e.g., K136[BPA]), suggesting an intimate functional relationship with U5-snRNP, reminiscent of hPrp28's role in facilitating U4/U6.U5 tri-snRNP integration into the

spliceosome[22,23]. Third, Prp28's contacts with Brr2 suggest that the two DExD/H-box ATPases physically communicate with each other for sequential removal of U1 snRNP and U4 snRNP. Two pre-B structures[24,25] are now available. In the yeast structure[24], Prp28 cannot be precisely positioned. Our cross-linking data, though, appears to fit well with both the yeast[24] and human[25] pre-B structures with respect to the locations of Prp8 and Snu114. In the human Pre-B structure, the main body of hPrp28 appears to be distant from the location of Brr2 (Supplementary Fig. 2c and Supplementary Software 1). Yet, the N-terminal domain of hPrp28, which is not conserved in yPrp28 (see below), threads through Prp8 and Snu114 moieties to reach Brr2 (Supplementary Fig. 2c). In both human and yeast cases, Brr2 is observed[24] or predicted[25], respectively, to undergo a rotation and trans-relocation in the pre-B-to-B transition[24,25]. In this light, we note that in the published yeast Pre-B structure, one of the speculated locations of Prp28 is close to Brr2 [24], while the other location is not; whereas the human Brr2, upon the predicted dramatic trans-relocation[25], would also be physically close to the Prp28 main body. It is therefore tempting to speculate that this trans-relocation may then place Brr2 in the vicinity of Prp28 main body that is made up of the two RecA-like (i.e., the enzymatic) domains. At the moment, we cannot rule out that the cross-linking between Prp28 and Brr2 can occur without translocation, because the contact can be through the N-terminal domain of Prp28. However, as the spliceosome complexes are highly dynamic during the splicing process, our biochemical approach might have captured a structurally dynamic, but so far undetected, intermediate state.

**Prp28's contacts with key spliceosomal proteins are functional.** To assess the physiological relevance of the Brr2 cross-linking data, we pairwise combined four cross-linking-compromised *prp28* alleles, each containing a 10 amino-acid deletion flanking the K82, K27, K41, or K136 residues (*prp28-K82Δ10, -K27Δ10, -K41Δ10, or -K136Δ10*), with seven *brr2* alleles. All double mutants exhibited synthetic-sick or -lethal phenotypes (Fig. 2a and Supplementary Fig. 5, Supplementary Table S1), suggesting that Prp28 and Brr2 are functionally interacting with each other. To analyze the physiological relevance of the observed cross-linking between Prp8 and Prp28, we performed a genome-wide genetic screen and uncovered *prp8-501*, a mutant allele of *PRP8* that resulted in a lethal phenotype in the Prp28-bypass background (*YHC1-1 prp28Δ*)[6]. This finding prompted us to test a panel of 47 other *prp8* alleles in a similar manner. Among them, 14 yielded lethality and one caused severe growth defect (Supplementary Data 6). The amino-acid changes inferred from all these mutant alleles are localized to a ~300-amino-acid region (1574–1883) partially overlapping with the maintenance of 3′SS fidelity region (1385–1625) and the structurally defined RNase H domain (1833–1950)[26]. Because all of these *prp8* alleles are recessive to the wild-type *PRP8* allele, the simplest interpretation would be that Prp8 acts in concert with Prp28 to promote U1 snRNP's departure. We note that genetic interaction between Prp28 and Snu114 was previously documented[27]. We next examined how these crosslink-compromised *prp28* alleles impact on spliceosome remodeling. Chromatin immunoprecipitation (ChIP) showed that U1 snRNP departure in the *prp28-K82Δ10* strain is delayed during co-transcriptional splicing of the *ACT1* pre-mRNA (Fig. 2b). Using an alternative in vitro strategy to monitor the U1 snRNP presence on assembled spliceosome, we found U1 snRNP accumulated at a higher level in *K27Δ10, K41Δ10, K82Δ10, and K136Δ10* reactions than that in the wild-type reaction and, importantly, the addition of purified Prp28 rescued this defect (Supplementary Fig. 6b, c).

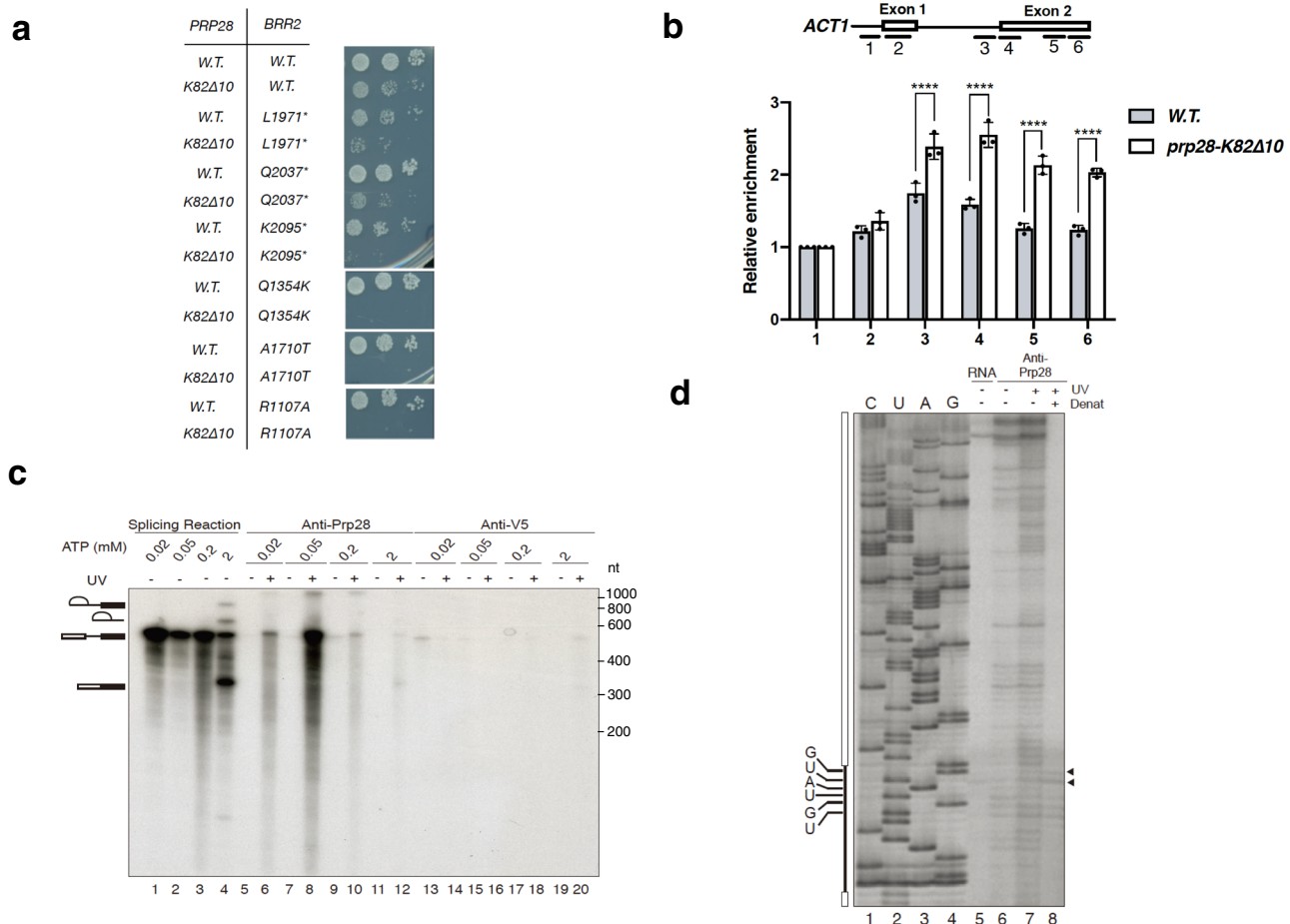

**Fig. 2 Prp28 genetically interacts with Brr2 and crosslinks to pre-mRNA 5′SS in the pre-B complex. a** Growth phenotypes of yeast strains harboring various combinations of *prp28-K82Δ10* and *brr2* alleles spotted in a dilution series to YPD medium at 30 °C. Asterisk represents stop codon. **b** ChIP analysis revealed that U1 snRNP's departure is delayed in the *prp28-K82Δ10* strain. Short lines in the top panel represent the amplicons used for monitoring U1 snRNP recruitment. In the bottom panel, the amplicon numbers are denoted on *X* axis. *Y* axis represents relative enrichment of U1 signal to that of the amplicon 1. Error bars are ± SEM; $n = 3$ biological repeats; ****$P < 0.0001$, unpaired two-tailed *t*-test. **c** Splicing reactions using radiolabeled *ACT1* transcript were done in wild-type extracts at 0.02, 0.05, 0.2, or 2 mM ATP and subjected to 254-nm UV irradiation (even-numbered lanes 6–20). Immunoprecipitations after denaturation were done with anti-Prp28 (lanes 5–12) or with negative-control anti-V5 antibody (lanes 13–20). Relative loadings are 1:1000 for splicing reactions alone (lanes 1–4) vs. immunoprecipitated reactions (lanes 5–20). The experiment was repeated three times with similar results. **d** Primer extension revealed two strong reverse transcriptase stops. Sequencing ladder (lanes 1–4) allows assignment of the two stops to the high conserved GU dinucleotide (lane 8, triangles) that remain after denaturation (Denat) and immunoprecipitation with anti-Prp28 antibody. Source data are provided as a Source Data file.

**Prp28 also contacts the conserved GU dinucleotide at the 5′ splice site**. To address whether Prp28 also makes specific contact with pre-mRNA, we irradiated the splicing reactions with UV (254 nm) to induce protein-RNA cross-linking. The cross-linked RNA was then immunoprecipitated under denaturing conditions. We found that Prp28's contact with pre-mRNA peaked at 0.05 mM ATP (Fig. 2c), consistent with the native immunoprecipitation data (Fig. 1a). Using oligonucleotide-directed RNase H cleavage followed by immunoprecipitation, we then assigned the contact point to a small region covered by oligonucleotides (oligos) D and E, the latter of which is downstream of 5′SS (Supplementary Fig. 7). To map the contact sites precisely, we use a downstream oligo for primer extension analysis. This analysis identified two strong reverse transcriptase stops at the highly conserved GU dinucleotide at the 5′SS under denaturing conditions (Fig. 2d). A previous experiment[28] using a different cross-linking method mapped hPrp28 contacts to predominantly + 7 position downstream of the 5′SS consensus. Taken together, Prp28 appears to behave as a conventional RNA helicase for unwinding the U1 snRNA/5′SS short duplex to dislodge U1 snRNP from the pre-B complex.

**Phosphorylated Npl3 potentiates Prp28's ATPase activity**. If Prp28 is indeed a canonical RNA helicase, a vexing question remains as to why it harbors only negligible ATPase activity. In our BPA experiments, we noticed Prp28-E326[BPA] yielded a particularly strong cross-linked product (Supplementary Fig. 9a, b). Mass-spectrometry analysis indicated that this protein is likely Npl3 (Supplementary Data 7 and Supplementary Figs. 8, 9c), a multifunctional SR-like RNA-binding protein involved in splicing[29] and mRNA export[30] and loosely associated with the purified U1 snRNP[29,31]. Immunoblotting analysis validated this cross-linked species as the phosphorylated form of Npl3 (p-Npl3) (Supplementary Figs. 9c and 10)[32]. Genetic analysis shows that the *prp28-E326Δ3 npl3Δ* double mutant exhibits a synthetic-lethal phenotype (Supplementary Fig. 9d). Immunoprecipitation analysis revealed that p-Npl3 associates with *ACT1* transcript in an ATP-independent fashion throughout the course of the splicing reaction (Fig. 3a). ChIP analysis reveals that U1 snRNP's recruitment was dramatically reduced in *npl3Δ* mutant (Fig. 3b and Supplementary Fig. 13), whereas in *prp28-E326Δ3* mutant U1 snRNP's departure is

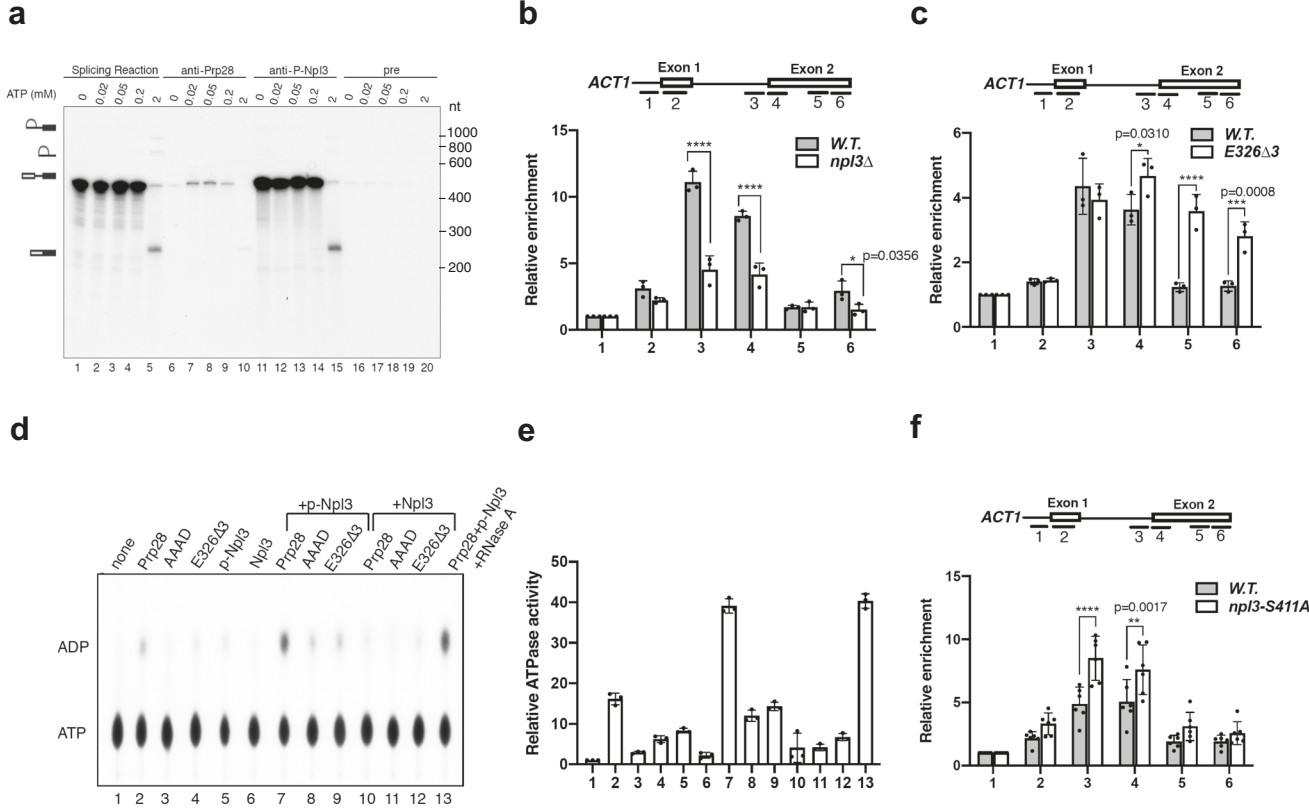

**Fig. 3 Npl3 is critical for U1 snRNP recruitment and phosphorylation of Npl3 activates Prp28's ATPase activity. a** p-Npl3 is associated with pre-mRNA in an ATP-independent manner. Splicing reactions (lanes 1–5) were done in splicing extracts at 0, 0.02, 0.05, 0.2, or 2 mM ATP and a portion was subjected to immunoprecipitation with either anti-Prp28 antibody (lanes 6–10), anti-p-Npl3 antibody (lanes 11–15), or pre-immune serum (lanes 16–20), respectively. Relative loadings are 1:10 for splicing reactions alone (lanes 1–5) vs. immunoprecipitated reactions (lanes 6–20). The experiment was repeated three times with similar results. **b** ChIP analysis showed that Npl3 is important for U1 snRNP recruitment. npl3Δ, npl3-deleted strain. Error bars are ± SEM; n = 3 biological repeats; ****P < 0.0001, *P < 0.05 (P = 0.0356), unpaired two-tailed t-test. **c** ChIP analysis revealed that U1 snRNP's departure is delayed in the prp28-E326Δ3 strain. Error bars are ± SEM.; n = 3 biological repeats; *P < 0.05 (P = 0.031), ****P < 0.0001, ***P < 0.001 (P = 0.0008), unpaired two-tailed t-test. **d, e** Prp28 ATPase assay. Purified Prp28, Prp28-AAAD, Prp28-E326Δ3, p-Npl3, and Npl3 were assayed for their ATPase activity in various combinations using [α-³²P]-ATP as a substrate. The products were separated by thin-layer chromatography for phosphoimager quantitation. Error bars are ± SEM; n = 3 biological repeats. **f** ChIP analysis showed that blocking S411 residue's phosphorylation in Npl3 (npl3-S411A) resulted in U1 snRNP accumulation in the spliceosome during co-transcriptional splicing. Error bars are ± SEM; n = 6 biological repeats; ****P < 0.0001, **P < 0.01 (P = 0.0017), unpaired two-tailed t-test. Source data are provided as a Source Data file.

significantly delayed (Fig. 3c). The fact that Prp28 lacks an N-terminal RS domain found in hPrp28, whose phosphorylation is critical for stable integration of U4/U6.U5 tri-snRNP into pre-B complex[22], raised a possibility that p-Npl3 is a missing cofactor for activating Prp28 ATPase. To test this hypothesis, we purified recombinant Prp28, Prp28-E326Δ3, Prp28-D341A/E342A (DEAD motif changed into AAAD)[10], Npl3, and p-Npl3 (Supplementary Fig. 9e) and examined their ATPase activities either alone or in various combinations (Fig. 3d, e). We found that only p-Npl3, but not Npl3, can promote Prp28's ATPase activity without the requirement of RNA addition (Fig. 3d, e). Critically, this ATPase activity is lost when Prp28-E326Δ3 is combined with p-Npl3. Finally, to assess the biological relevance of Npl3 phosphorylation at S411, which is phosphorylated by Sky1[32], we performed ChIP analysis using the npl3-S411A mutant strain. U1 snRNP was found to co-transcriptionally accumulate at a higher level in npl3-S411A than that in the wild-type strain (Fig. 3f). We, therefore, suggest that p-Npl3 is a cofactor that stimulates Prp28's ATPase activity to remove U1 snRNP from the pre-B complex. However, the precise nature, such as the exact phosphorylation site (s), of such p-Npl3 within the splicing complexes remains to be determined.

**Npl3 may correspond to the metazoan-specific N-terminal region of Prp28.** The hPrp28 has an N-terminal region, consisting of an RS domain (residues 1–221) and an anchor domain (residues 286–356). A recent human Pre-B complex structure[25] showed that this anchor domain, which is immediately N-terminal to the RecA1 domain, docks at the interface between Prp8 N terminus and Snu114 (Supplementary Fig. 2c). Intriguingly, although this anchor domain is well conserved in metazoans but it is completely absent in yeast (Supplementary Fig. 2). This leaves a puzzle as to how Prp28 is recruited to the yeast Pre-B complex. The fact that Prp28 physically interacts with Npl3 during splicing (Supplementary Fig. 8c and Supplementary Data 7) and that Npl3 also contains an RS domain suggest that Npl3 may be a functional counterpart of the hPrp28 N-terminal region. In this light, we suggest a working model (Fig. 4) in which p-Npl3 may correspond to the human RS and anchor domains and that its phosphorylation regulates Prp28's ATPase activity. The latter is especially important, because an inopportune ATP hydrolysis may negatively impact on the reported Prp28's proofreading activity[33]. Our finding that phosphorylation at S411 activates Npl3 to promote Prp28's ATPase activity also raises a broader implication that a similar strategy may apply to other DEAD/H-box helicases for spatial and temporal regulation within

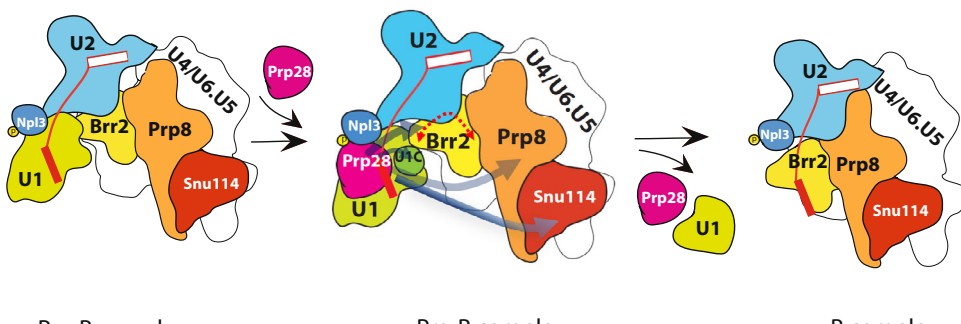

**Fig. 4 Model for Prp28's recruitment to the pre-B complex and its ATPase activation by p-Npl3.** The yeast pre-B complex (left panel) contains U1 (blue), U2 (green), U4/U4.U6 (white) snRNPs, which harbors Brr2 (yellow), Prp8 (orange) and Snu114 (red) proteins. In this diagram, U1, U2, U4/U4.U6, Brr2, Prp8, and Snu114 are drawn to scale, shape, and orientation following the published yeast pre-B structure. In this structure, p-Npl3 (deep blue and a yellow circle, which stands for phosphorylation) is already located at a site very close to the intron (thin red line) 5'SS. Exon 1 (filled red rectangular box) and exon 2 (empty rectangular box) are not drawn to the scale. In the middle diagram, Prp28 is recruited by p-Npl3 into the pre-B complex and then physically contacts U1C, Brr2, Prp8, and Snu114, which may serve to orient Prp28 at the right time and the right place so that the p-Npl3-activated Prp28 can hydrolyze ATP to expel U1 snRNP, resulting in the formation of U1-free B complex (right panel). The double-headed thick dash line (in red) above Brr2 (middle panel) depicts the reported and speculated Brr2 relocation, which may bring Brr2 to the vicinity of Prp28 for direct contact, thereby signaling Brr2 to remove U4 snRNP leading to the B$^{act}$ complex (not shown). Finally, we note that p-Npl3 was found to remain bound to the later splicing complexes and also to the spliced mRNA.

a complex environment. Finally, our data show that, through two different regulatory routes, the function and behaviors of Prp28 and hPrp28 within the spliceosome are fundamentally kept in evolution.

## Methods

**Yeast strains**. Strains used in this study are described in Supplementary Data 1.

**Plasmids and oligonucleotides**. Plasmids and oligonucleotides used in this study are described in Supplementary Data 2 and 3, respectively.

**Antibodies and reagents**. The following antibodies were used in this study: anti-Prp8 (ref.[34]; 1:2000), anti-Snu114 (ref.[34]; 1:2000), and anti-Brr2 antibodies. All three antibodies were gifts from Dr. Soo-Chen Cheng (Institute of Molecular Biology, Academia Sinica, Taiwan). Note that the anti-Brr2 was raised against a His-tagged Brr2 N-terminal fragment [aa 1–183] and is highly robust and specific (at 1:2000 dilution) in detecting Brr2 in the yeast extract by immunoblotting or by immunoprecipitation. The anti-p-Npl3 and anti-Npl3 were obtained from Dr. Christine Guthrie (UCSF, USA) (ref.[35]; 1:2000). Note that anti-p-Npl3 detects only p-Npl3, but not Npl3; anti-Npl3 detects both phosphorylated and non-phosphorylated Npl3. Anti-Prp40 was a gift from Dr. Paul Siciliano, University of Minnesota (ref.[36]; 1:2000). Anti-Prp28 was raised in this lab (ref.[37]; 1:2000). In addition, the following reagents were from commercial sources: *p*-Benzoyl-L-phe-nylalanine (Bpa) (Bachem), anti-HA.11 (ms, Cat # MMS-101R, Lot # B220850, Covance, Clone: 16B12, 1:2000), anti-V5-TAG (ms, Cat # MCA1360, Lot # 0915, Bio-Rad, Clone: SV5-PK1, 1:2000), anti-Maltose Binding Protein (MBP) (ms, Cat # E8032S, Lot # 0101603, NEB, Clone: B48, 1:10,000), anti-GAPDH (ms, Cat # G8795-200, Lot # 045M4799V, Sigma, 1:10000), HRP-conjugated anti-rabbit IgG (H + L) (gt, Cat # 65–6120, Lot # QK229568, Invitrogen, 1:10,000–1:40,000), HRP-conjugated anti-mouse IgG (H+L) (gt, Cat # 62–6520, Lot # QG215721, Invitrogen, 1:10,000~1:40,000), Immobilon Western Chemiluminescent HRP Substrate (Millipore), SuperSignal West Femto Maximum Sensitivity Substrate (Thermo), Polyvinylidene difluoride membrane (Amersham$^{TM}$ Hybond 0.45 mm PVDF, GE), Protein A-Sepharose (PAS) was obtained from GE Healthcare Life Sciences, Protein A Mag Sepharose Xtra (PAmS) (GE), Amylose Resin (New England Biolabs), 5-Fluoroorotic acid (5-FOA) (ZYMO RESEARCH), Proteinase K was purchased from MD Bio Inc., and Fast SYBR Green Master Mix (Applied Biosystems), Perfect RNA marker template mix (0.1–1 kb, Novagen, Cat # 69003-3).

**Splicing extracts, radioactively labeled RNA, and splicing assays**. Yeast whole-cell extracts were prepared according to the liquid-nitrogen-grinding method[38]. Actin precursor RNA substrates were synthesized in vitro as runoff transcripts using SP6 RNA polymerase and labeled with [α-$^{32}$P] UTP at 20 Ci/mmole, which was defined as 1× specific activity for the *ACT1* pre-mRNA substrate. The 10× specific activity was defined as substrates labeled with [α-$^{32}$P] UTP at 200 Ci/mmole. The full-length transcript is purified from 5% polyacrylamide (acrylamide-bisacrylamide [29:1]/8 M urea) gels. Standard splicing assays[39] were carried out for 30 min at 25 °C unless otherwise indicated.

**Phosphorylated Npl3 add-back experiments**. Two reactions (70 μl each), using either wild-type or *npl3Δ* splicing extract, were assembled and incubated. Aliquots (10 μl) were withdrawn at 5, 10, 15, 20, and 30 min and treated with proteinase K to stop the reaction for analyzing the splicing progression. Because at 5 min, a distinctive difference was observed between the two reactions, we, therefore, chose the 5-min incubation time for subsequent add-back experiments. To test whether p-Npl3 can rescue the delay of splicing in the *npl3Δ* reaction, purified p-Npl3 (see below) was added into the reaction mix without ATP and pre-incubated for 10 min. ATP was then added to start the splicing reaction for a total of 5 min.

**Co-Immunoprecipitation assay**. Protein A Mag Sepharoses Xtra (10 μl) were pre-bound with 1 μl of polyclonal anti-p-Npl3 or with 2 μl of polyclonal anti-Npl3 complemented with 500-μl NET2 buffer (150 mM NaCl, 50 mM Tris-HCl [pH 7.4], 0.05% NP40), which was then placed on a rotating platform for 1–2 h at 4 °C. After binding, the resin was washed three times with 1-ml ice-cold NET2 buffer and stand on the ice. p-Npl3 (1.2 μg) or Npl3 (1.2 μg) was mixed with Prp28 (1.2 μg) under splicing condition without RNA transcript, incubated at 25 °C for 30 min, and then mixed with IgG-bound beads at 4 °C for 1 h. Beads were washed with 1-ml cold NET2 three times. Bound proteins were separated by 4–20% gradient gel for probing with either anti-Prp28, anti-p-Npl3, or anti-Npl3 antibody. In the reciprocal experiments, anti-Prp28 antibody (2 μl) was bound to the beads.

**Immunoblotting**. Standard immunoblotting analysis was used to detect Prp28 and its covalently cross-linked products. In a typical Western procedure, anti-Prp28 polyclonal antibody (1:2000 dilution) and HRP-conjugated anti-rabbit IgG (1:10,000 or 1:40,000 dilution, depending on the HRP substrate used subsequently) were employed. Alternatively, mouse monoclonal anti-HA11 antibody (1:2000 dilution) and HRP-conjugated anti-mouse IgG (1:10,000 or 1:40,000 dilution) were used. PVDF blotting membranes were used. Immobilon Western Chemiluminescent HRP Substrate was chosen for developing the Western blot when the secondary antibody was used in 1:10,000 dilution. For more sensitive detection, SuperSignal West Femto Maximum Sensitivity Substrate was employed along with 1:40,000 dilution of the secondary antibody. Fujifilm Super RX-N Blue X-Ray Film was used for detecting chemiluminescence. The exposure time was typically from 1 to 10 min. For evaluation of Prp28 levels in wild-type and *prp28* "small-deletion" mutants, we picked two independent colonies for growth in 1 l YPD medium at 30 °C to 3–4 OD$_{600}$. Yeast extracts were prepared by the liquid-nitrogen-grinding method[38]. Equal protein amounts (20–30 μg) were loaded for SDS-PAGE, followed by transfer to PVDF membranes, and for immunoblotting analysis. The same method was used to examine Npl3 levels in the wild-type and the *npl3Δ* strains. Anti-Npl3 and anti-p-Npl3 polyclonal antibodies (1:2000 dilution) and anti-GAPDH monoclonal antibody (1:10,000 dilution) were used.

**BPA-mediated cross-linking experiments**. The BPA-cross-linking method has been previously described in a step-by-step manner[17], which includes cell growth in the presence of BPA, extract preparation, spliceosome assembly on the MS2 loop-tagged *ACT1* transcript, UV (365 nm) irradiation, spliceosome pull down, denaturation of pull-down spliceosome, SDS-PAGE, and detection of the cross-linked species by specific antibodies. This detailed protocol[17] also includes the scale-up procedure for mass-spectrometry analysis.

**Detection of spliceosome-associated Prp28 by RNA pull-down experiment**. Method for using MS2 loop-tagged transcript to pull down assembled spliceosome has been previously described in a step-by-step manner[17], except, in this case, a splicing extract containing V5-tagged Prp28 was used and neither BPA incorporation nor UV irradiation was employed. Prp28 was detected by an anti-V5 antibody.

**Mass-spectrometry analysis**. After silver-staining the gel, regions corresponding to the locations of the cross-linked product in both lanes (with or without UV irradiation) were excised and subjected to standard mass-spectrometry analysis (LC–MS/MS). The purified recombinant protein (1.2 µg) of p-Npl3 was resolved by SDS-PAGE and Coomassie-Blue stained, then the single band was retrieved and analyzed for phosphorylation position by MS analysis (Supplementary Fig 10d). The phosphopeptide MS analysis presented was done from the same batch of phosphorylated Npl3 that was used in the ATP hydrolysis assay and in the in vitro co-affinity studies. Samples were detected by Nano-LC-NanoESI-MS/MS on an Orbitrap Fusion mass spectrometer (ThermoFisher Scientific, San Jose, CA) equipped with EASY-nLC 1200 system (Thermo, San Jose, CA, US) and EASY-spray source (Thermo, San Jose, CA, US). The gradient solution was injected (5 µl) at 1 µl/min flow rate on to easy column (C18, 0.075 mm × 150 mm, ID 3 µm; Thermo Scientific). Chromatographic separation was using 0.1% formic acid in water as mobile phase A and 0.1% formic acid in 80% acetonitrile as mobile phase B operated at 300 nl/min flow rate. Briefly, the gradient employed was 2% buffer B at 2 min to 40% buffer B at 40 min. Full-scan MS condition: mass range $m/z$ 375–1800 (AGC target 5E5) with lock mass, resolution 60,000 at $m/z$ 200, and maximum injection time of 50 ms. The MS/MS was run in top speed mode with 3-s cycles with CID for protein id or ETD for phosphopeptide; while the dynamic exclusion duration was set to 60 s with 10 ppm tolerance around the selected precursor and its isotopes. Electrospray voltage was maintained at 1.8 kV and the capillary temperature was set at 275 °C. Data analysis was analyzed as follows: Raw file processed by Maxquant (Version 1.6.14.0). MS/MS spectra were searched against the UniProt yeast database (UP000002311, 6049 entries) and enzyme digestion by trypsin with 2 missed cleavage sites. The parameter such as MS tolerance of 20 ppm for the first search and main search 6 ppm. Fragment tolerance was 0.5 Da (IT) or 0.01 Da (FT). Variable modification of oxidation (M) and acetylation (protein N-terminal) and fixed modification of carbamidomethyl (C) as search parameters. To evaluate enrichment of the protein upon UV cross-linking, we adopted the LFQ (Label-Free Quantification) intensity of protein for quantification[40,41]. In addition, variable modification of phospho (STY) for the phosphopeptide identification was analyzed and the localization probabilities were filtered larger than 0.95 (Supplementary Data 8). LC–MS/MS Quantification of site-specific phosphorylation degree of p-Npl3 was done by Xcalibur Qual Browser 4.1.31.9 (Supplementary Fig. 11). The mass-spectrometry experiments were done by two biological repeats.

**Immunoprecipitation of the spliceosome**. To detect the physical association of Prp28, p-Npl3, or Prp40 with spliceosome, we first assembled spliceosome using [$^{32}$P]-labeled *ACT1* transcript. Immunoprecipitation of spliceosome was performed as described[42]. In these experiments, 1-µl anti-Prp28, 0.1-µl anti-p-Npl3, or 1-µl anti-Prp40 antibody was pre-incubated with protein A-Sepharose. The splicing reaction (10 µl) was then incubated with 10-µl antibody-coupled protein A-Sepharose (~2.5 mg) for precipitating the spliceosome. The precipitates were washed with 1-ml cold NET-2 three times at room temperature. After proteinase K treatment (1 mg/ml for 37 °C/30 min), RNAs were extracted for urea poly-acrylamide (8%) gel electrophoresis.

**Chromatin immunoprecipitation (ChIP) analysis**. ChIP analysis was done exactly as described[7], except that anti-V5 (U1C-V5) and anti-Prp40 were used to precipitate the chromatin-bound U1 snRNP. Briefly, 50 ml of ~0.8 OD$_{600}$ yeast cells were treated with formaldehyde (2%). Next, cells were resuspended in lysis buffer and disrupted by bead beater. The insoluble cross-linked fraction was subjected to sonication for shearing the DNA fragment to ~300 bp. Subsequently, immuno-precipitation of U1C-V5 or Prp40 was performed by anti-V5-PAS or anti-Prp40-PAS beads at 4 °C overnight. After extensive washes, crosslinks were reversed and DNA extracted. Primer pairs (Supplementary Table S3) covering the *ACT1* gene were used in qPCR experiments. All data represent an average of three independent experiments. Error bars are ±SEM. The enrichment of the U1C-bound or Prp40-bound chromatin over the background was normalized to that of primer set 1 data points.

**Genetic screen of the *prp8-501* mutation**. YTC302 (*MAT**a** prp28::HIS3 YHC1-1 ade2 ade3* pCA8105-*PRP28/ADE3/URA3*) were grown in YPD (1% yeast extract, 2% bacto-peptone and 2% glucose), diluted and plated to YPD containing 4% glucose (4% YPD). Plates were exposed to 254-nm UV to obtain an 80–90% killing rate and incubated at 30 °C. Colonies that did not appear to the sector were picked and re-streaked twice to 4% YPD. Those remaining non-sectoring were tested on media containing 1 mg/ml 5-FOA to select against cells containing the *URA3* gene. 5-FOA-sensitive clones were transformed with pCA8034-*PRP28/LEU2*. Candidates able to sector and become 5-FOA resistant were tested for reversion of the *YHC1-1*

locus by introducing pYHC1003-*YHC1-1/LEU2*. Those that remained 5-FOA sensitive were isolated. One strain that met all the criteria is YTC303 (*MAT**a** prp28::HIS3 YHC1-1 prp8-501* pCA8105). This was crossed to a non-mutagenized isogenic strain YTC301 (*MATα prp28::HIS3 YHC1-1 PRP8* pCA8105-*PRP28/ADE3/URA3*). Sectoring and viability on 5-FOA were restored, indicating the mutation is recessive. Tetrad analysis of the resulting diploid yielded 2:2 segregation for survival on 5-FOA indicating that a single gene had been mutated. YTC303 was then transformed with a wild-type YCp50-based (*LEU2*-marked) genomics library. Transformants that regained the ability to sector were chosen and the library plasmid was recovered and sequenced. The library clone contained only the *PRP8* open reading frame, revealing that a mutation in *PRP8* was synthetically lethal with *prp28Δ YHC1-1*. The library plasmid containing wild-type *PRP8* is pPRP8001 (YCp50-*PRP8*). The region of *PRP8* containing the synthetic-lethal mutation was identified by gap repair using pPRP8002-*PRP8/TRP1* that was gapped separately with three enzymes, *AflII, AgeI*, and *BstEII*. Linearized gapped plasmids were then transformed into YTC303. Only the plasmid that was gapped with *BstEII* plasmid failed to provide relief from synthetic lethality. This plasmid was rescued and sequenced to identify *prp8-501* and was named pPRP8003. Mutant alleles of *prp8* were obtained from other laboratories (Supplementary Table 7). Plasmids carrying the mutant alleles of *PRP8* were tested for synthetic lethality with *prp28Δ YHC1-1* by introduction into YTC307 (*prp28::HIS3 YHC1-1 prp8::LYS2* pJU169-*PRP8/URA3*). Transformation of *prp8* mutant alleles was followed by selection on media containing 1 mg/ml 5-FOA. Plasmids that conferred synthetic lethality were rescued from the yeast strain and sequenced to verify the mutation within *PRP8* that caused lethality. To ensure that the *PRP8* alleles causing lethality on 5-FOA were synthetically lethal and not null mutations, plasmids were transformed into YJU75 (*prp8::LYS2* pJU169-*PRP8/URA3*) and the transformants were found to be 5-FOA viable.

**Genetic analysis of *prp28, brr2*, and *npl3***. To construct the *prp28* tester strains, we individually transformed plasmids pRS415-*prp28-K27Δ10*, pRS415-*prp28-K41Δ10*, pRS415-*prp28-K82Δ10*, or pRS415-*prp28-K136Δ10* into YTC1282 and counter-selected pRS316-*PRP28* by plating the cells on SC-Leu/5-FOA plate. The *prp28* tester strains were crossed with strain YTC1446 (*MAT**a**, brr2::natMX4* pRS416-*BRR2*) and the diploid cells were sporulated. We selected the haploid testers with *prp28Δ::kanMX4*, pRS415-*prp28* alleles, *brr2::natMX4*, and pRS416-*BRR2*. The haploid tester strains were further transformed with pRS413-*BRR2*, pRS413-*brr2-A1710T*, pRS413-*brr2-G1708\**, pRS413-*brr2-K2095\**, pRS413-*brr2-L1971\**, pRS413-*brr2-Q2037\**, pRS413-*brr2-Q1354K*, pRS413-*brr2-R1107A*[43], respectively. The transformants were spotted on SC-His/-Leu/5-FOA plates. The inviable or slow growth cells indicated synthetic lethality or sickness of tested *prp28* alleles and tested *brr2* alleles. The *prp28* tester strain (*prp28Δ::kanMX4*, pRS415-*prp28-E326Δ3*) was mated with the *npl3* deletion strain (BY4741 *MAT**a**, npl3::natMX4*) and the resulting diploids were sporulated. We selected the haploid containing *prp28Δ::kanMX4*, pRS415-*prp28-E326Δ3*, *npl3::natMX4* for spotting assay.

**UV cross-linking of Prp28 to pre-mRNA**. Splicing reactions (25 µl each) were carried out in a 12-well culture plate with 10X specific activity of *ACT1* pre-mRNA substrate (2 nM) for 30 min, quenched on ice, and then UV-irradiated (254 nm; 0.8 J/cm$^2$) on ice in a CL-1000 Ultraviolet Crosslinker (UVP). The UV light source is placed ~5 cm above the sample. UV-irradiated samples were denatured in 1% SDS (w/v), 1% Triton X-100 (v/v), and 100 mM DTT, heated in boiling water for 90 sec, and then placed at 25 °C for 2 min. The denatured samples were centrifuged in a microfuge at 16,363 × $g$ for 1 min at room temperature. The supernatants were removed to new tubes. Before immunoprecipitation, the concentration of denaturants was reduced by 10-fold dilution with NET-2–300 (300 mM NaCl, 50 mM Tris-HCl [pH 7.4], 0.05% NP40) and tRNA added at a final concentration of 200 µg/ml. The mixtures were incubated with 25 µl of antibody-conjugated protein A Sepharose (PAS) for 1.5 h at 4 °C. The precipitates were washed with 1 ml of cold NET-2–300 five times and 1 ml of room temperature NET-2 (150 mM NaCl, 50 mM Tris-HCl [pH 7.4], 0.05% NP40) twice. Finally, after proteinase K treatment (1 mg/ml for 37 °C/30 min), RNAs were extracted and dissolved each in 10 µl formamide dye for denaturing urea gel (8%) electrophoresis.

**Oligonucleotide-directed RNase H and primer extension experiments**. A 70-µl splicing reaction containing 2 nM of radiolabeled (10× specific activity; 200 Ci/mmol) *M3-ACT1* pre-mRNA and 0.05 mM ATP was first assembled, from which an aliquot of 1 µl was removed as "untreated" control. The remaining mix was treated with 5 µM oligonucleotide in a total of 73 µl reaction for 20 min at 25 °C, from which an aliquot of 1 µl was removed as an oligonucleotide-treated control. From the remaining volume, a 10-µl aliquot was removed and incubated with anti-Prp28-PAS for 1.5 h at 4 °C as an immunoprecipitation control. The rest of 60 µl was then split into two halves, one of which was irradiated with UV and the other was not. Both were then denatured, treated, and immunoprecipitated as described above. RNAs from untreated control and oligonucleotide-treated control were extracted and dissolved in 200 and 20 µl formamide dye, respectively. The RNA from immunoprecipitated control was extracted and dissolved in 20 µl formamide dye. The RNAs from denatured reactions were extracted and each dissolved in 5 µl

formamide dye. An equal volume (5 µl) from all those samples was then analyzed by denaturing urea gel (8%) electrophoresis. To map Prp28's contact site on pre-mRNA, we scaled up the reaction markedly to 60 ml. All the steps before the primer extension step are the same as described for the RNase H experiment. For primer extension analysis, RNA was isolated from 60 ml of splicing reaction mixture after UV irradiation. Each RNA sample was mixed with a 5′-end [$^{32}$P]-labeled oligonucleotide #6 ($4 \times 10^5$ cpm), denatured by heating at 65 °C for 5 min, and quickly chilled on ice. Primer extension reactions were carried out with RevertAid H Minus First Strand cDNA Synthesis Kit (Thermo) at 42 °C for 1 h. DNA products were recovered after degrading the RNA by NaOH and fractionated on an 8% sequencing gel (acrylamide-bisacrylamide [19:1]/8 M urea) and visualized by autoradiography. The same primer was used to generate the sequencing ladder using pSPACT1 as a template. Sequencing reactions were done by using Sequence$^{TM}$ Version 2.0 DNA Sequencing Kit (Affymetrix).

**Purification of recombinant Prp28$^{WT}$ and Prp28$^{AAAD}$ proteins**. We typically grew 6 l of *E. coli* culture for purifying recombinant Prp28$^{WT}$ or Prp28$^{AAAD}$ protein. Two plasmids were created under the pRSET-A (Addgene) vector backbone carrying wild-type yeast Prp28$^{WT}$ and its mutant Prp28$^{AAAD}$ protein-coding regions in frame with N-terminal 6-His tag. *E. coli* Rosetta cells (Stratagene) were employed and protein expression was induced by adding 0.3 mM IPTG at 37 °C for 3 h. Cells were collected by centrifugation and resuspended in a buffer containing 50 mM KH$_2$PO$_4$, 50 mM Na$_2$HPO$_4$ (pH 7.4), 500 mM NaCl, 3 mM β-mercaptoethanol, 7.5% (v/v) glycerol, supplemented with complete EDTA-free protease inhibitor cocktail (Roche). The cells were harvested and lysed with a microfluidizer and the lysate was centrifuged in a JA25.50 rotor (Beckmann Coulter) at 34,957 × g for 40 min at 4 °C. Then, the supernatant was mixed with Ni-Sepharose 6 Superflow affinity resins (Qiagen) for 30 min at 4 °C followed by a stepwise prewash with buffer contains 25 mM and 50 mM imidazole. After that, Prp28p$^{WT}$ and Prp28p$^{AAAD}$ proteins were eluted with buffer containing 250 mM imidazole. Eluted proteins were pooled and dialyzed for 16 h at 4 °C in a buffer containing 20 mM HEPES (pH 7.4), 150 mM NaCl, 3 mM dithiothreitol, and 7.5% (v/v) glycerol. After dialysis, protein samples were loaded into MonoS 5/50 GL column (GE Healthcare Life Sciences), equilibrated in a buffer containing 20 mM HEPES (pH 7.4), 150 mM NaCl, 3 mM dithiothreitol, and 7.5% (v/v) glycerol, and bound proteins were eluted with a linear gradient of NaCl concentration to 1 M. Protein-containing fractions were loaded onto a HiLoad 200 10/300 GL column (GE Healthcare Life Sciences) equilibrated in a buffer containing 20 mM HEPES (pH 7.4), 150 mM NaCl, 3 mM dithiothreitol and 7.5% (v/v) glycerol. Proteins fractionated (0.5 ml) from HiLoad 200 10/300 GL column (GE Healthcare Life Sciences) were collected and analyzed by Coomassie-stained SDS-PAGE (12%) and for mass-spectrometry analysis. These two proteins were concentrated to 0.4 mg/ml in 20 mM HEPES (pH 7.9), 0.2 mM EDTA, 50 mM KCl, 20% glycerol, 0.5 mM DTT and stored at −80 °C.

**Purification of recombinant p-Npl3 proteins**. We typically grew 10 l of *E. coli* culture for purifying recombinant p-Npl3 or Npl3 protein. Phosphorylation of Npl3 was carried out in Rosetta cells by co-expressing Npl3 and Sky1. Full-length DNA fragments of Npl3 and Sky1 protein-coding genes were PCR-amplified and cloned, respectively, into *NdeI/XhoI* sites of a modified pET28a (Novagen) vector with a PreScission protease site directly after the N-terminal 6-His tag and into *BamHI/XhoI* sites of pGEX-6p vector (GE Healthcare) directly after N-terminal GST tag. Clones were verified by DNA sequencing. Protein expression was induced by the addition of 0.5 mM IPTG at 18 °C for 10 h. Cells were collected by centrifugation and resuspended in a buffer containing 50 mM KH$_2$PO$_4$, 50 mM Na$_2$HPO$_4$ (pH 7.4), 500 mM NaCl, 3 mM β-mercaptoethanol, supplemented with a complete EDTA-free protease inhibitor cocktail (Roche). Cells were harvested and lysed with a microfluidizer and the lysate was centrifuged in a JA25.50 rotor (Beckmann Coulter) at 34,957 × g for 40 min at 4 °C. The supernatant was then mixed with Ni-NTA Superflow affinity resin (Qiagen) for 40 min at 4 °C followed by a stepwise prewashed with buffer containing 25 mM and 50 mM imidazole. Npl3 was eluted by a linear gradient of imidazole concentration to 500 mM. Eluted proteins were pooled and dialyzed for 16 h at 4 °C with a buffer containing 20 mM HEPES (pH 7.4), 500 mM NaCl, 2.5% (v/v) glycerol, and 3 mM dithiothreitol. After dialysis, protein samples were loaded onto MonoQ 5/50 GL column (GE Healthcare Life Sciences), fractions containing flow-through were pooled and further purified over a Superdex 200 10/300 GL column (GE Healthcare Life Sciences) equilibrated in a buffer containing 20 mM HEPES (pH 7.4), 500 mM NaCl, 2.5% (v/v) glycerol and 3 mM dithiothreitol. Peak fractions (0.5 ml) were collected and analyzed by Coomassie-stained 12% SDS-PAGE. Likewise, non-phosphorylated Npl3 was purified by the same method without co-expressing Sky1. These two proteins were concentrated to 0.4 mg/ml in 20 mM HEPES (pH 7.9), 0.2 mM EDTA, 50 mM KCl, 20% glycerol, 0.5 mM DTT and stored at −80 °C.

**ATPase assay**. Before assembling the reaction, p-Npl3 or Npl3 was pre-mixed with Prp28 and incubated on ice for 30 min. The reaction (10 µl) contains 0.57 µM Prp28 (wild type or AAAD mutant), with or without 1.6 µM p-Npl3 (or Npl3), together with 0.2 µl of [α-$^{32}$P]ATP (3,000 Ci/mmol; Perkin-Elmer), 40 µM cold ATP, 40 mM Tris-HCl (pH 8), 40 mM KCl, 1.6 mM MgCl$_2$, 0.08 mg/ml bovine

serum albumin, and 0.8 mM dithiothreitol. The reaction was incubated at 37 °C for 2.5 h and stopped with 4 µl of 100 mM EDTA. 2 µl of the reaction was spotted onto a TLC (PEI) plate, which was developed with a buffer containing 0.375 M potassium phosphate at pH 3.5 for 1.5 h to separate ADP and ATP. The amount of ATP hydrolyzed was quantified using a PhosphorImager on a Typhoon scanner (GE Healthcare).

**Statistical analysis and software**. The cryo-EM structure of the labeled human pre-B complex (Protein Data Bank 6QX9) was displayed by PyMol 2.3.2 (Supplementary Fig. 2c and Supplementary Software 1). Gels and TLC (PEI) plate were exposed to GE Storage Phosphor Screen (GE Healthcare Life Sciences). RNA bands and the amount of ATP hydrolyzed were quantified with Typhoon$^{TM}$ FLA9000 (GE Healthcare Life Sciences) and analyzed with ImageQuant TL7.0 (GE Healthcare Life Sciences). All statistics were performed using GraphPad Prism 8 (Graphpad). Data are presented as the mean ± SEM. Statistical significance between the two groups was determined with an unpaired two-tailed Student's *t*-test. ChIP experiments were carried out with *n* = 3 biological repeats (Figs. 2b, 3b, c, and Supplementary Fig. 1c), and *n* = 6 biological repeats (Fig. 3f). Prp28 ATPase assay was carried out with *n* = 3 biological repeats (Fig. 3e). Quantification of retained U1 snRNP from spliceosome was carried out (Supplementary Fig. 6b), *n* = 3 biological repeats.

**Reporting summary**. Further information on research design is available in the Nature Research Reporting Summary linked to this article.

## Data availability

The raw mass-spectrometry proteomics data have been deposited to the ProteomeXchange Consortium via the PRIDE partner repository with the data set identifiers PXD024492 (Prp28-K136BPA)[44]; PXD024493 (Prp28-E326BPA)[45]; PXD024494 (p-Npl3)[46]. The uncropped gel or blot figures and original data underlying Figs. 1–3 and Supplementary Figs. 1–13 are provided as a Source Data file[47]. All data supporting the findings of this study are available from the corresponding authors upon reasonable request. Source data are provided with this paper.

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

## Acknowledgements

We thank C. Guthrie and M. Konarska for providing *prp8* alleles; C. Guthrie for anti-p-Npl3 and anti-Npl3 antibodies; P. Siliciano for anti-Prp40 antibody; S.-C. Cheng for anti-Prp8, anti-Brr2, and anti-Snu114 antibodies and advice on primer extension analysis; H.-T. Chen for BPA-related plasmids and reagents; S.-W.E. Chen and M.-D. Tsai for assistance in mass-spectrometry analysis. T.-H.C was supported by the Ministry of Science and Technology (MOST 105–2311-B-001–059 and 109-2311-B-001-035), Academia Sinica Thematic Project grant (AS-103-TP-B12), and Academia Sinica (AS-SUMMIT-109) and Ministry of Science and Technology (AS-KPQ-109-BioMed) and MOST 109-0210-01-18-02.

## Author contributions

T.-H.C. conceived and designed the project; F.-L.Y., S.-L.C., G.R.A., L.T., H.-I.L, C.-S.Y., W.-Y.H., L.S.L., C.M., C.-M.L., and S.-C.T. conducted the experiments; T.-H.C., F.-L.Y. S.-L.C., and W.-H.C. analyzed the data; F.-L.Y., S.-L.C., W.-H.C., and T.-H.C. wrote the paper.

## Competing interests

The authors declare no competing interests.
