## [Peer Review File · Nature Communications]

Reviewers' comments:

Reviewer #1 (Remarks to the Author):

Prp28 is a DEAD-box helicase that plays an important role in pre-mRNA splicing and is partly conserved from yeast to humans. This protein is essential for release of U1snRNP from the pre-mRNA and for progression of the splicing reaction. Despite the recent cryo-electron microscopy structures, the recruitment, regulation and mechanism of action of yeast Prp28 remain unclear because, unlike in the human spliceosome structures, yeast Prp28 is disordered in most EM structures.

In this study, the authors biochemically investigate the association of Prp28 with different spliceosomal proteins in course of the splicing cycle using BPA-driven cross-linking. The authors make use of an orthogonal aminoacyl tRNA synthetase-suppressor tRNA system to incorporate the photoactivable unnatural amino acid BPA at specific sites of Prp28 and prepared splicing extracts from yeast cells expressing these engineered proteins. Cross-linking in combination with mass-spectrometry and antibody-dependent detection of known splicing factors performed under different ATP concentrations allowed the authors to identify U1C, Prp8, Brr2 and Snu114 as interactors of Prp28. The interactions with U1C were previously reported; those with Prp8, Brr2 and Snu114 suggest an association of yeast Prp28 with U5 snRNP, reminiscent of the human protein. Disruption of the Prp28-Brr2 interaction leads to severe growth defects and synthetic lethality in yeast. Furthermore, consistent with its role in U1snRNP release, specific Prp28 mutants lead to delay in U1 snRNP departure in during co-transcriptional splicing. Finally, the authors show that the yeast SR-like protein Npl3p, upon phosphorylation by the kinase Sky1p, associates strongly with Prp28 and enhances its ATPase activity. This leads to the model that Npl3p in yeast might have a functional role equivalent to the N-terminal RS domain of human Prp28.

The authors present an interesting biochemical study of the transient interactions mediated by Prp28, which provides valuable insight into the function of the yeast protein. Major concerns and additional comments about this study are listed below.

Major concerns:

- One concern regarding the growth defect experiments performed in yeast is that deletion of 10 residues within the protein (as in Prp28 K82delta10) might lead to production of a construct that is simply not well-behaved and viable in yeast. To eliminate this possibility, it is recommended that the authors demonstrate that the deletion mutants used in the yeast growth assays are expressed at the same (or comparable) levels as the wild-type protein.
- The association of phospho-Npl3p with Prp28 is a major finding of the paper. Given that the authors have the purified proteins on hand, demonstrating a direct interaction between phospho-Npl3p and Prp28 (either by pulldown or size-exclusion chromatography) would greatly strengthen the manuscript.
- It would be very helpful if the authors commented on the interactions mediated by human Prp28 with proteins in the tri-snRNP or other spliceosomal factors so that one is in a position to compare the human and yeast systems while reading through the results. Similarly, a more extensive discussion highlighting the similarities and unique features of yeast Prp28 in comparison to the human homolog and a speculation into how Npl3p might influence ATPase activity (or indeed how the RS domain of human Prp28 modulates its ATPase activity) would make for a more interesting manuscript.

Additional comments:

- Page 3, line 30 – "..., which is disassembled to recycle..." should read "..., which is disassembled to recycle...".
- The authors should mention, within the main text, how the sites of BPA-incorporation were selected.
- Although it is fairly easy to understand this, it would be helpful if the authors expanded the legend of Supplementary figure 3 to describe what wt, 5'SS, 3'SS, Δ-in etc stand for.
- Are the data in Figures 2b, 3b and 3c the average (mean) of the 3 biological replicates? This should

be specified in the figure legend.

- It is important to verify phosphorylation of the purified Npl3p co-expressed with Sky1 by mass-spectrometry, particularly to substantiate the claim that only phospho-Npl3p activates Prp28.

Reviewer #2 (Remarks to the Author):

In the Manuscript titled "Activation of Prp28 ATPase by Phosphorylated Npl3 at a Critical Step of Spliceosome Remodeling", Yeh et al. use site-specific labeling of Prp28 to capture a transient spliceosomal intermediate that reveals which other proteins Prp28 interacts with in the spliceosome to promote release of the U1 snRNP in an ATP-dependent manner. As the authors state, understanding how ATPases regulate and trigger conformational changes in RNPs remains a difficult challenge, not just for RNA splicing, but all aspects of RNA biology.

To identify transient Prp28 interactions with other proteins and the pre-mRNA during release of the U1 snRNP from the 5'SS, the authors placed a photo-activatable unnatural amino acid (BPA) in various positions of Prp28 in vivo, and prepared splicing extracts, within which they could crosslink and capture Prp28 in action with UV light. The authors provide evidence that Prp28 crosslinks with the conserved 5'SS GU dinucleotide and the tri-snRNP proteins Prp8, Brr2, and Snu114. They support these physical interaction observations with genetic interaction studies. They also find that Prp28 binds a phosphorylated form of Npl3 that promotes Prp28 ATPase activity and may be a functional counterpart of the N-terminal RS domain found in human Prp28 but lacking in yeast. Overall the data are solid and exhaustive, but are interpreted optimistically to support an overall model for detailed action of this important helicase.

While the authors propose a very interesting model and have a strong record discovery in this area, the current work needs significant reorganization in order to bring it to a state where it can be digested readily by readers not intimately familiar with the details of splicing. Several interpretations need to be modified, for example, the data do not show that Prp28 can simultaneously contact Prp8, Brr2, and Snu114 (line 79-80). It is possible there are three (or more) complexes within which only one of the proteins contacts Prp28 at a time. Genetic results such as a synthetic lethal interaction with recessive *prp8* alleles is taken as evidence that "Prp8 acts synergistically with Prp28 to promote U1 snRNP's departure" (Line 104), when other alternatives seem equally positive (eg additivity vs synergy, or other models that are not excluded). The numerous pieces of evidence – crosslinking – genetic – cell biological – are not interwoven in an organized way and rather at certain points the topic seems to change, and readers will get lost. An example is the paragraph starting on line 90 which would seem to be about Brr2, but after two sentences brings in the incompletely explained genetic interactions with Prp8. These organizational issues will lose general readers and make the paper hard to follow. It is a sophisticated set of well-executed experiments and the findings are important but the presentation needs work. The format seems exceedingly compressed, and there are many supplemental figures and tables, not all of which may be necessary, but some of which might be better in the main document. Constant reference to the supplemental items is required at a number of points in order to follow the logic, and this makes it hard to read.

General comments:

The authors should provide more detail on how the crosslinking and mass spec was performed (number of replicates, immunoprecipitation, washing, etc). It would be nice to know the relative enrichment of identified proteins in no UV vs UV conditions, as it is hard to evaluate signal to noise in several places.

Chromatin immunoprecipitation of U1 across the actin gene seems a poor way to make the mechanistic point that Prp28 release is delayed by mutations or the absence of a protein like Npl3. Is there no biochemical assay for this? In addition, no splicing extracts or biochemical experiments with

the Npl3-S113A mutant are shown. It seems to make the mechanistic point suggested by the genetics that such extracts or Npl3 deletion extracts with add back should be done. Would be much stronger.

Specific comments follow.

Figure 1:

- In Figure 1A, Prp28 co-immunoprecipitates mRNA and intron lariat at 2 mM ATP. Do the authors believe Prp28 is still weakly associated with late-stage splicing complexes? In other words it seems that the spliced product and some lariat are recovered – is this background?
- The left panel of Figure 1E does not show Prp28 release as indicated by the authors in lines 47-49.
- 1C: Have the authors tested if incorporation of BPA into Prp28 affects splicing in vitro? BPA incorporation might inhibit Prp28 activity and thus splicing kinetics.
- 1C: As a control, the authors should probe for proteins identified by mass spec but not likely directly crosslinked to Prp28. Rrp5, Tho2, GpH1 for example as a negative control.
- In Figure 1B (and later in Figure 2C), pre-mRNA associated with Prp28 peaks at 0.05 mM ATP. However, in 1C crosslinking for K27, K41, and K82 seems to peak at 0.2 and 2 mM. Why does this differ? Do the authors think this reflects different conformations of Prp28?
- Related: On lines 79-80, the authors conclude that Prp28 can simultaneously contact Prp8, Brr2, and Snu114. While the data does suggest that Prp28 interacts with these proteins, it does not discriminate between these interactions occurring individually over time or simultaneously. This should be reworded to clarify.

Supplementary Figure 3:

- It is unclear what the labels (5'SS, BP, and 3'SS) mean for the gel in Figure S3A. Do these mean these sequences are deleted? Mutated? The authors describe these substrates in legend for Supplemental Figure S8, but this should be moved to the legend of S3 as well.
- On lines 61-63, the authors state that crosslinked species are splicing dependent, because their appearances depend on the presence of pre-mRNA, intron, functional 5'SS and branch site. However, Snu114 is detected in K82 (b) and K136 (c) in the absence of a functional branch site and 3'SS. How do the authors reconcile this? Do they believe it represents assembly but not catalysis?

Supplementary Table 3:

- The use of BPA to crosslink and capture Prp28 in a transient splicing complex is one of the successes of this manuscript. Why not put some version of table S3 describing crosslinked proteins by mass spec in the main manuscript.? Perhaps the top 10 most enriched proteins upon UV crosslinking.
- It is unclear to the reviewer what the "number of peptides assigned" means in no UV vs UV conditions. Is this supposed to convey enrichment of the protein upon UV crosslinking? If so, it is not convincing and contradicts the data in Supplementary Figures S3 and S4 that suggests crosslinking is required to capture Brr2 (13 peptides vs 16), Prp8 (13 vs 19), etc. There are a number of (likely) unrelated proteins that seem to have similar numbers of peptides compared to spliceosomal proteins claimed to crosslink to Prp28. For example, Rrp5, Tho2, Gph1. Also: big proteins (like Prp8 and Brr2) have more peptides – how was this normalized? What is the abundance of these proteins in the

starting extract? Abundant proteins may give more peptides depending on the sensitivity.

If possible, the authors should provide the relative fold enrichment of peptides assigned in UV compared to no UV conditions.

Furthermore, the authors should state the number of replicates as shown in supplementary table S6

- It would be very useful for the reader to provide the standard name of the protein in addition to the systematic name.

Supplementary Figure S5:

- On lines 93-95 the authors state that prp28-136 Δ 10 brr2-G1708 and prp28-136 Δ 10 brr2-L1971 manifested in a suppressive interaction at 30 °C. However, in Supplementary Figure S5, both double mutants grow (or fail to grow) worse than either single mutant. This should be corrected or clarified.

Supplementary Figure S7:

- The nomenclature for the oligos used for RNase H digestion in S7 is confusing with the amplicon numbering of ChIP amplicons in Figure 2B. Maybe one could use letters instead of numbers.

Supplementary Table S6:

- The comments above about Supplementary Table S3 are also relevant here. It is unclear that UV crosslinking enriches for direct interactions. For example, Npl3 shows only 3 peptides in each of two replicates with UV crosslinking but no peptides without crosslinking. This is substantially lower than some of the other proteins listed in Table S6 and S3. Again, in addition to the number of identified peptides, this should be portrayed as fold enrichment upon UV crosslinking.

Miscellaneous concerns:

- Overall, the Methods section seems to be lacking sufficient detail for a reader to reproduce many of the experiments.
- It would be useful to know at what dilution primary antibodies were used at, as well as what secondary antibodies were used for detection.
- The Prp28-HA strains are not described in the Methods section
- The "BPA-mediated crosslinking experiments" and "Immunoprecipitation of the spliceosome" sections of the Methods should be expanded to at least provide a brief summary of the experiments performed.
- There are no methods describing how mass spec was performed and how the data was analyzed.
- There is also very little indication of the number of replicates for each mass spec experiment.
- In lines 32-39 that describe the role of Prp28 in releasing U1 snRNA, it seems important to reference original work from Staley and Guthrie, 1999.
- Lines 102-103 carry a reference of a Prp8 region (1817-1976) as the "uridine-tract recognition domain" (Umen et al 1996). The name for this region should be checked against the available cryoEM structures and perhaps a different name should be used if in fact the indicated region is not near the uridine tract.

Reviewer #3 (Remarks to the Author):

The manuscript from Yeh et al. examines the mechanism of activation of the ATPase activity of a DEAD-box protein, Prp28, that catalyzes a key step in spliceosome activation: transfer of the intron 5' splice site from pairing with U1 snRNA to pairing with U6 snRNA. Because the ATPase activity of purified Prp28 is very low, it is suspected that Prp28 is activated only in the context of the spliceosome pre-B complex, in which all five snRNPs are present and the 5' splice site is still paired with U1. The authors use a site-specific protein-protein crosslinking approach to make the striking discovery that Prp28 is activated by phospho-Npl3 (p-Npl3), which is not a known constituent of the spliceosome but, as an hnRNP protein, is likely bound to the pre-mRNA. The combination of in vitro and in vivo experiments done by the authors to validate the involvement of p-Npl3 in spliceosome formation and activation is generally quite convincing, although the case would be strengthened by experiments analyzing in vitro splicing using extracts from a Sky1 or Npl3 deletion strain supplemented with either Npl3 or p-Npl3. The combination of the exhaustive Prp28 crosslinking data and the intriguing, albeit somewhat preliminary, identification of p-Npl3 as an activator of Prp28 make this manuscript a significant contribution to the gene expression field. However, several issues enumerated below require attention.

1) Supplementary Figure 2 shows the location of BPA substitution in the crystal structure of purified Prp28, which is not very informative. Instead, the sites of BPA should be shown in the primary structure of Prp28 annotated with the location of domains and helicase motifs, AND in the cryo-EM structure of the human pre-B complex (PDB 6QX9; ref. 5). The former will reveal how the substitutions may affect ATPase function and the latter will show which interactions the crosslinks are expected to capture. It is not possible to map these in the yeast pre-B complex because Prp28 is not resolved in that structure. Fortunately, Prp28 is sufficiently conserved that most if not all of the BPA sites should be identifiable in the human structure. The annotated cryo-EM structure should also be supplied as a PyMOL file in supplementary material.

2) In Supplementary Tables 3 (which lists the Prp28-K136(BPA)-crosslinked proteins identified by mass spec.) and 6 (which lists Prp28-E326(BPA)-crosslinked proteins identified by mass spec.) the common names of the proteins should be supplied in addition to the systematic names, when they exist.

3) In line 69 is Prp28 meant rather than Prp8?

4) The authors should attempt to explain the strong crosslink between Prp28-K41(BPA) and Brr2 at 2 mM ATP, using the human pre-B structure or structures from later in the splicing cycle. Figure 4 is inadequate in this regard since Brr2 is far from Prp28 in every panel. BPA is essentially a zero Angstrom crosslinker so near-atomic structures should be used to rationalize the results. It is not clear what the word "trans-relocation" on line 87 means.

5) There is either a labeling or interpretation problem with Supplementary Figure 5 (right). The authors claim that two double mutants, prp28-K136del10 with either brr2-G1708* or -L1971*, manifest a suppressive interaction at 30°C (line 94). However, the double mutant with G1708* clearly grows slower than either single mutant, and the double mutant with L1971* is dead!

6) I disagree that a synthetic lethal phenotype supports the interactions identified, which is stated on line 95 and implied on line 133. If either mutation A or mutation B ablates an interaction, then the combination of the two should have no greater effect.

7) The Figure legend for Supplementary Figure 6 that starts at line 358 does not agree with the legend printed on the figure. The legend on the figure correctly states that the Y axis shows the amount of U1 snRNP detected (not "departed") and that 100% is set by the wild-type reaction at 0.05 mM (not "2 mM"). In general, there should not be two legends for a single figure in a submitted manuscript. This is the case for all the supplementary figures.

8) Line 145: "this ATPase activity is lost when Prp28-E326del3 is combined with p-Npl3". "Lost" should be replaced with "diminished" since the ATPase activity in lane 9 is greater than that in lanes 4 or 12 of Figure 3d/e. Are these differences significant?

9) The authors should define "NET-2" solution since there are different formulations on the web.

10) Line 492: "10x specific activity" is not useful if 1x is not defined.

11) If mass spectrometry of BPA-cross linked proteins is described in ref.30, that should be stated. If not, it should be referenced or described.

REVIEWER #1:**Major Concerns**

- (1) One concern regarding the growth defect experiments performed in yeast is that deletion of 10 residues within the protein (as in Prp28 K82delta10) might lead to production of a construct that is simply not well-behaved and viable in yeast. To eliminate this possibility, it is recommended that the authors **demonstrate that the deletion mutants used in the yeast growth assays are expressed at the same (or comparable) levels as the wild-type protein.**

As suggested, we performed immunoblotting assay for assessing the steady-state levels of Prp28 in the wild-type strain and in “Δ10” mutant strains (K27, K41, K82, and K136). There appear no significant differences of Prp28 levels (two biological repeats) among these strains. These data are now shown in **Supplementary Fig. 5b**. The experimental procedure is described in great details in the revised Methods under the “Immunoblotting” section.

- (2) The association of phospho-Npl3p with Prp28 is a major finding of the paper. Given that the authors have the purified proteins on hand, **demonstrating a direct interaction between phospho-Npl3p and Prp28 (either by pulldown or size-exclusion chromatography) would greatly strengthen the manuscript.**

Using purified Prp28 and Npl3p, we performed reciprocal co-IP experiments, which shows that p-Npl3 interacts with Prp28 more favorably than Npl3 under *in vitro* condition. These data are now displayed in **Supplementary Fig. 10**. The experimental procedure is described in the revised Methods under the “Co-Immunoprecipitation assay” section.

We wish to note that monitoring 2-component interaction between Prp28 and p-Npl3 *in vitro* may not entirely reflect their interaction within the highly complex spliceosome environment. In this sense, our BPA crosslinking, mass-spec, and later antibody probing data have provided powerful supports for the initial mass-spec data.

- (3) It would be very helpful if the authors **commented on the interactions mediated by human Prp28 with proteins in the tri-snRNP or other spliceosomal factors** so that one is in a position to compare the human and yeast systems while reading through the results. Similarly, **a more extensive discussion highlighting the similarities and unique features of yeast Prp28 in comparison to the human homolog and a speculation into how Npl3p might influence ATPase activity** (or indeed how the RS domain of human Prp28 modulates its ATPase activity) would make for a more interesting manuscript.

In human, Prp28 (hPrp28) is stably associated with the pre-B spliceosomal complex (Teigelkamp, S. *et al.* [1997] *RNA* **3**, 1313-1326; Boesler, C. *et al.* [2016] *Nat. Commun.* **7**, 1197-1208). In this context, Prp28 is physically close to Prp8, Brr2, and Snu114, all U5-snRNP components, and U1C (Charenton, C. *et al.* [2019] *Science* **364**, 362-367). Here, we showed that, in yeast, the yPrp28 also interacts with Prp8, Brr2, Snu114, and U1C during splicing. The hPrp28 has a N-terminal region, consisting of an RS domain (residues 1–221)

and an anchor region (residues 286–356), the latter of which is responsible for anchoring hPrp28 to pre-B complex (Charenton, C. *et al.* [2019] *Science* 364, 362-367). However, this N-terminal region is completely missing in yPrp28. The fact that the yPrp28 interacts with Npl3 during splicing thus raises a possibility that Npl3 may be a functional surrogate of the hPrp28 N-terminal domain, because Npl3 contains an RS domain. Thus, the yPrp28 may be anchored to spliceosome, much like the scenario in the human system.

Structures of the hPrp28 (residues 352–806) and yPrp28 (residues 127–588) have been solved. Both structures exhibit a wide-open conformation between two RecA domains, presumably resulting in a low ATPase activity (Mohlmann, S. *et al.* [2014] *Acta Crystallogr., Sect D: Biol. Crystallogr.* 70, 1622-1630; Jacewicz, A. *et al.* [2014] *Nucleic Acids Res.* 42, 12885-12898). Phosphorylation of the hPrp28 RS domain is critical for integration of the U4/U6-U5 tri-snRNP into the spliceosome (Mathew, R. *et al.* [2008] *Nat. Struct. Mol. Biol.* 15, 435-443). We speculate that this phosphorylation may then activate hPrp28's ATPase activity, thereby moving the splicing reaction forward. In this work, we showed that the phosphorylated form of Npl3 (p-Npl3) is capable of promoting yPrp28 ATPase activity. Perhaps the phosphorylation event, be it within the hPrp28 RS domain or in the yNpl3, represents an evolutionarily conserved process, through which Prp28 activity is potentiated. As a support, we further showed that p-Npl3 enhances *in vitro* splicing activity (**Supplementary Fig. 11b**) and is sufficient to reverse the deficient splicing activity of *npl3Δ* extracts (**Supplementary Fig. 11c**).

The expanded explanation is now included in the revised main text marked in red (see page 9, lines 191–201).

Additional Comments:

- (1) Page 3, line 30 –“..., which is dissembled to recycle...” should read “..., which is **disassembled** to recycle...”.

Thanks for picking up the typo. We have corrected the error accordingly.

- (2) The authors should mention, within the main text, how the sites of BPA-incorporation were selected.

As suggested, we have added the following paragraph in the main text, highlighted in red in the revised manuscript (see page 4, lines 84–90).

“There are several considerations for choosing which amino-acid residues for BPA replacement. We first selected hydrophilic amino-acid residues that are not strictly conserved among DExD/H-box proteins, arguing that they are more likely to situate on Prp28p's surface and that their BPA replacements are less likely to significantly impact on Prp28p's function. We then used the structural information of Vasa (Sengoku, T. *et al.* [2006] *Cell* 125, 287-300), another DExD/H-box protein, to computationally model Prp28p structure and to guide our final selections.”

- (3) Although it is fairly easy to understand this, it would be helpful if the authors expanded the legend of Supplementary figure 3 to describe what wt, 5'SS, 3'SS, Δ -in etc stand for.

As suggested, we have expanded the legend of Supplementary Figure 3 to describe what wt, 5'SS, 3'SS, Δ -in are.

- (4) Are the data in Figures 2b, 3b and 3c the average (mean) of the 3 biological replicates? This should be specified in the figure legend.

Yes, these data are average of 3 biological replicates. We have added the sentence: "Error bars \pm s.e.m.; n=3" to the figure legend.

- (5) It is important to verify phosphorylation of the purified Npl3p co-expressed with Sky1 by mass-spectrometry, particularly to substantiate the claim that only phosphor-Npl3p activates Prp28.

As suggested, we have mapped the phosphorylated sites, by mass-spectrometry, of the purified p-Npl3 (co-expressed with Sky1 in *E. coli*). The mass-spectrometry data have been deposited to the ProteomeXchange repository (PRIDE), and the result was presented in Supplementary Table 9. The data showed that there are twenty phosphorylated residues in the C-terminal region of p-Npl3 (amino acids 306–411), including the anticipated Ser411, the mutation of which impacts on the U1 snRNP (Fig. 4).

REVIEWER #2:

Overall Comments (*excerpt*): Overall the data are solid and exhaustive, but are interpreted optimistically to support an overall model for detailed action of this important helicase. While the authors propose a very interesting model and have a strong record discovery in this area, the current work needs significant reorganization in order to bring it to a state where it can be digested readily by readers not intimately familiar with the details of splicing. **Several interpretations need to be modified, for example, the data do not show that Prp28 can simultaneously contact Prp8, Brr2, and Snu114 (line 79-80). It is possible there are three (or more) complexes within which only one of the proteins contacts Prp28 at a time.**

We have toned down the sentence, which now reads as:

"Second, Prp28 can contact Prp8, Brr2, and Snu114 (e.g., K136^{BPA}), suggesting an intimate functional relationship with U5 snRNP, reminiscent of hPrp28's role in facilitating U4/U6.U5 tri-snRNP integration into the spliceosome." (Page 5, Lines 115–117)

Genetic results such as a synthetic lethal interaction with recessive *prp8* alleles is taken as evidence that **"Prp8 acts synergistically with Prp28 to promote U1 snRNP's departure"**

(Line 104), when other alternatives seem equally positive (e.g. additivity vs synergy, or other models that are not excluded).

Agree. We toned down the sentence, without using “synergistically”, as follows:

“Prp8 acts in concert with Prp28 to promote U1 snRNP’s departure”. (see Page 7 and Line 144).

The numerous pieces of evidence – crosslinking – genetic – cell biological – are not interwoven in an organized way and rather at certain points the topic seems to change, and readers will get lost. An example is the paragraph starting on line 90 which would seem to be about Brr2, but after two sentences brings in the incompletely explained genetic interactions with Prp8. These organizational issues will lose general readers and make the paper hard to follow. It is a sophisticated set of well-executed experiments and the findings are important but the presentation needs work. **The format seems exceedingly compressed, and there are many supplemental figures and tables, not all of which may be necessary, but some of which might be better in the main document.** Constant reference to the supplemental items is required at a number of points in order to follow the logic, and this makes it hard to read.

We wish to note that this manuscript was originally written for and submitted to *Nature*, so the format by necessity was highly compressed. This manuscript was later recommended by *Nature* Editor for direct transferring to *Nature Communications*. As a result, the format remained unchanged. We have taken reviewer’s comments seriously by carefully revising and expanding in this R1 version, so that general readers can more easily follow the story. The revised and expanded portions within the R1 main text are now specifically marked in red for reviewer’s tracking.

General Comments

- (1) The authors should provide more detail on how the crosslinking and mass spec was performed (number of replicates, immunoprecipitation, washing, etc). It would be nice to know the relative enrichment of identified proteins in no UV vs UV conditions, as it is hard to evaluate signal to noise in several places.

As recommended, we have revised the “BPA-mediated crosslinking experiments” and “Mass spectrometry analysis” sections in Methods, which describe how experiments and quantitative analysis were performed. Relative enrichment is now shown in the modified **Supplementary Table 5**. Experiments for detecting crosslinked species by Western were repeated at least 3 times. Mass-spectrometry analysis for identifying the crosslinked species was repeated twice each.

- (2) Chromatin immunoprecipitation of U1 across the actin gene seems a poor way to make the mechanistic point that Prp28 release is delayed by mutations or the absence of a protein like Npl3. Is there no biochemical assay for this? In addition, no splicing extracts or biochemical experiments with the Npl3-S113A mutant are shown. It seems to make the mechanistic point suggested by the genetics that such extracts or Npl3 deletion extracts with add back should be done. Would be much stronger.

In the splicing field, ChIP analysis is conventionally and routinely used for detecting snRNPs' association with newly synthesized RNA transcripts. Note that we also employed an alternative method by immunoprecipitating spliceosome-associated U1 snRNP and then quantitated the levels of the spliceosome-associated hot transcript as a readout (see Supplementary Figure 6b).

Following Reviewer's comment, we performed the add-back experiments. As expected, p-Npl3, but not unphosphorylated Npl3, enhances *in vitro* splicing activity and is sufficient to rescue the loss of splicing activity of *npl3Δ* extracts (Supplementary Fig. 11).

Specific Comments:

Figure 1:

- (1) In Figure 1A, Prp28 co-immunoprecipitates mRNA and intron lariat at 2 mM ATP. Do the authors believe Prp28 is still weakly associated with late-stage splicing complexes? In other words, it seems that the spliced product and some lariat are recovered - is this background?

For the following reasons, we believe that Prp28 remains, at least in part, associated with the late-stage splicing complexes:

- (1) The data shown in Fig. 1a are highly reproducible in numerous experiments;
- (2) ChIP analysis following Prp28's association with the newly synthesized transcript clearly showed that the Prp28 association remains for the last primer-set probe corresponding to the 3' end of the *ACT1* gene (Supplementary Figure 1cd). It is known that by the time the 3' end is made, splicing reaction would have been completed (Oesterreich, F. C. *et al.* [2016] *Cell* 165, 372-381). Thus, a fraction of Prp28 is most likely still associated with the late-stage splicing complexes.
- (2) The left panel of Figure 1E does not show Prp28 release as indicated by the authors in lines 47-49.

As suggested, we have added a diagram of Prp28 release on the right-hand side of Figure 1e.

- (3) 1C: Have the authors tested if incorporation of BPA into Prp28 affects splicing *in vitro*? BPA incorporation might inhibit Prp28 activity and thus splicing kinetics.

As suggested, we have done the *in vitro* splicing assays for K27, K41, K82, and K136 extracts (Supplementary Fig. 1e), which show that here is no significant impact on splicing activity in these Prp28-BAP-containing extracts.

- (4) 1C: As a control, the authors should probe for proteins identified by mass spec but not likely directly crosslinked to Prp28. Rrp5, Tho2, GpH1 for example as a negative control.

To address this issue, we repeated the Prp28-K136 (BPA)-crosslinking experiment, which detected Prp8 as a crosslinked species upon UV irradiation. The same membrane was then stripped off signals and re-probed with anti-Rrp5 antibody, i.e., Rrp5 was used as a negative control. As shown in Supplementary Fig. 1f,g, there is no enrichment of crosslinked species of Rrp5 upon UV irradiation, as expected.

- (5) In Figure 1A (and later in Figure 2C), pre-mRNA associated with Prp28 peaks at 0.05 mM ATP. However, in 1C crosslinking for K27, K41, and K82 seems to peak at 0.2 and 2 mM. Why does this differ? Do the authors think reflects different conformations of Prp28?

We wish to note that experiments shown in Fig. 1A and 2C employed the classical “bulk” biochemical approaches, in which the “averaged-out” amount of Prp28 associated with a mixture of splicing complexes at various stages was examined.

In contrast, experiments described in Fig. 1B were to examine the Prp28-(individual protein) interactions, again in a mixture of splicing complexes at various stages enriched (but not arrested) by using various ATP concentrations. This approach, despite still being a “bulk” one, bears an element of the “single-molecule” approach.

As a result, it is possible that Prp28 interacts with different proteins at different “time” (roughly marked by different ATP concentrations) among a mixture of splicing complexes. This interpretation is consistent with what we have already addressed in the preceding Q1: “...Prp28 is most likely still associated with the late-stage splicing complexes”.

- (6) Related: On lines 79-80, the authors conclude that Prp28 can simultaneously contact Prp8, Brr2, and Snu114. While the data does suggest that Prp28 interacts with these proteins, it does not discriminate between these interactions occurring individually over time or simultaneously. This should be reworded to clarify.

We agree with the reviewer. As explained in Q5, Prp28 may interact independently with either of the three proteins at different stages of splicing complexes, i.e., the interaction of Prp28 with these three proteins needs not to be “simultaneous”.

Supplementary Figure 3:

- (1) It is unclear what the labels (5’SS, BP, and 3’SS) mean for the gel in Figure S3A. Do these mean these sequences are deleted? Mutated? The authors describe these substrates in legend for Supplemental Figure S8, but this should be moved to the legend of S3 as well.

We updated the figure legends for Supplementary Figures S3a and S8 by adding clear definition of 5’SS, BP, and 3’SS. In short, they mean transcripts containing specific mutations at these sites.

- (2) On lines 61-63, the authors state that crosslinked species are splicing dependent, because their appearances depend on the presence of pre-mRNA, intron, functional 5'SS and branch site. **However, Snu114 is detected in K82(b) and K136(C) in the absence of a functional branch site and 3'SS.** How do the authors reconcile this? Do they believe it represents assembly but not catalysis?

Yes, the weakly detected Snu114-crosslinked species is perhaps more related to spliceosome assembly, but not to the catalysis *per se*.

<On the branch-site issue> The transcript we used in the reaction contains a well-known “pseudo-branch-site”, UACUAAG, which inefficiently functions as a branch site, upon losing the genuine wild-type branch site (UACUAAC) (Cellini, A. *et al.* [1986] *Mol. Cell. Biol.* 6, 1571-1578; Vijayraghavan, U. *et al.* [1986] *EMBO J.* 5, 1683-1695). Therefore, a minor fraction of spliceosome might have been assembled on the branch-site mutant transcript, offering a plausible explanation for the weak crosslinking of Snu114 to Prp28.

<On the 3'SS issue> It is well known that the 3'SS-mutated transcript (i.e., the so-called ACAC transcript) still allows spliceosome to assemble for completing the first splicing step (Vijayraghavan, U. *et al.* [1986] *EMBO J.* 5, 1683-1695). This may explain, once again, the weak crosslinking of Snu114 and Prp28.

Supplementary Table 3:

- (1) The use of BPA to crosslink and capture Prp28 in a transient splicing complex is one of the successes of this manuscript. Why not put some version of table S3 describing crosslinked proteins by mass spec in the main manuscript? Perhaps the top 10 most enriched proteins upon UV crosslinking.

As recommended, we have added a Top 10 List (**Supplementary Table S5b**). We chose to keep this List in the Supplementary for avoid sidetracking the flow of the story.

- (2) It is unclear to the reviewer what the “number of peptides assigned” means in no UV vs UV conditions. Is this supposed to convey enrichment of the protein upon UV crosslinking? If so, it is not convincing and contradicts the data in Supplementary Figures S3 and S4 that suggests crosslinking is required to capture Brr2 (13 peptides vs 16), Prp8 (13 vs 19), etc. There are a number of (likely) unrelated proteins that seem to have similar numbers of peptides compared to spliceosomal proteins claimed to crosslink to Prp28. For example, Rrp5, Tho2, Gph1. Also: big proteins (like Prp8 and Brr2) have more peptides-how was this normalized? What is the abundance of these proteins in the starting extract? Abundant proteins may give more peptides depending on the sensitivity. If possible, the authors should provide the relative fold enrichment of peptides assigned in UV compared to no UV conditions. Furthermore, the authors should state the number of replicates as shown in supplementary table S6.

For clarity, we have re-configured the presentation of **Supplementary Table S5** and Supplementary Figure S4. To normalize the data, we followed the published method widely accepted in the mass-spectrometry field by using LFQ (Label-free Quantification) (Cox, J. *et al.* [2014] *Mol Cell Proteomics* 13, 2513-2526). The calculated folds of enrichment are now clearly shown in the Table. Note that those splicing-dependent crosslinked proteins we have pursued have all been independently validated by Western analysis.

- (3) It would be very useful for the reader to provide the standard name of the protein in addition to the systematic name.

This is now done, as now shown in **Supplementary Table S5**.

Supplementary Figure S5:

On lines 93-95 the authors state that *prp28-136Δ10 brr2-G1708* and *prp28-136Δ10 brr2-L1971* manifested in a suppressive interaction at 30°C. However, in Supplementary Figure S5, both double mutants grow (or fail to grow) worse than either single mutant. This should be corrected or clarified.

We apologize for this glaring error. With no exception, all pairs of tested mutations resulted in either synthetic lethality or sickness. We now deleted the mis-statement from the relevant sentence from the main text.

Supplementary Figure S7:

The nomenclature for the oligos used for RNase H digestion in S7 is confusing with the amplicon numbering of CHIP amplicons in Figure 2B. Maybe one could use letters instead of numbers.

As recommended, we have changed the name of oligos from numbers to letters (A–E) in **Supplementary Figure S7**.

Supplementary Table S6:

The comments above about Supplementary Table S3 are also relevant here. It is unclear that UV crosslinking enriches for direct interactions. For example, Npl3 shows only 3 peptides in each of two replicates with UV crosslinking but no peptides without crosslinking. This is substantially lower than some of the other proteins listed in Table S6 and S3. Again, in addition to the number of identified peptides, this should be portrayed as fold enrichment upon UV crosslinking.

We have done the re-calculation of the Npl3 mass-spec data and re-configured its presentation. The calculated folds of enrichment are now clearly shown in the **Supplementary Table S8**. Note that those splicing-dependent Npl3 has been extensively validated by Western analysis using anti-Npl3 antibody.

Miscellaneous concerns:

We shall address **Q1-Q6** altogether below, as they all deal with adding small technical details to the R1 version.

- (1) Overall, the Methods section seems to be lacking sufficient detail for a reader to reproduce many of the experiments;
- (2) It would be useful to know at what dilution primary antibodies were used at, as well as what secondary antibodies were used for detection;
- (3) The Prp28-HA strain are not described in the Methods section;
- (4) The “BPA-mediated crosslinking experiments” and “Immunoprecipitation of the spliceosome” sections of the Methods should be expanded to at least provide a brief summary of the experiments performed;
- (5) There are no methods described how mass spec was performed and how the data was analyzed;
- (6) There is also very little indication of the number of replicates for each mass spec experiment.

All the relevant information related to **Q1-Q6** is now included in the expanded Methods.

- (7) In lines 32-39 that describe the role of Prp28 in release U1 snRNA, it seems important to reference original work from Staley and Guthrie, 1999.

The said reference has been added to the expanded reference list.

- (8) Lines 102-103 carry a reference of a Prp8 region (1817-1976) as the “**uridine-tract recognition domain**” (Umen et al 1996). The name for **this region should be checked against the available croEM structures and perhaps a different name should be used** if in fact the indicated region is not near the uridine tract.

As suggested, we now rewrite the sentence as follows:

“The amino-acid changes inferred from all these mutant alleles are localized to a ~300-amino-acid region (1574–1883) partially overlapping with the maintenance of 3’SS fidelity region (1385–1625) and the structurally defined RNase H domain (1833–1950).”

REVIEWER#3:

Overall Comments (*excerpt*): The combination of *in vitro* and *in vivo* experiments done by the authors to validate the involvement of p-Npl3 in spliceosome formation and activation is generally quite convincing, although the case would be strengthened by experiments analyzing *in vitro* splicing using extracts from a Sky1 or Npl3 deletion strain supplemented with either Npl3

or p-Npl3. The combination of the exhaustive Prp28 crosslinking data and the intriguing, albeit somewhat preliminary, **identification of p-Npl3 as an activator of Prp28 make this manuscript a significant contribution to the gene expression field. However, several issues enumerated below require attention.**

- (1) Supplementary Figure 2 shows the location of BPA substitution in the crystal structure of purified Prp28, which is not very informative. **Instead, the sites of BPA should be shown in the primary structure of Prp28 annotated with the location of domains and helicase motifs, AND in the cryo-EM structure of the human pre-B complex (PDB 6QX9; ref. 5).** The former will reveal how the substitutions may affect ATPase function and the latter will show which interactions the crosslinks are expected to capture. It is not possible to map these in the yeast pre-B complex because Prp28 is not resolved in that structure. Fortunately, Prp28 is sufficiently conserved that most if not all of the BPA sites should be identifiable in the human structure. **The annotated cryo-EM structure should also be supplied as a PyMOL file in supplementary material.**

We have substantially revised **Supplementary Figure 2** to address all the requested information:

- (1) Marking BPA sites on the yPrp28 linear sequence, on which various helicase motifs are also marked as well;
 - (2) Sequence alignment of yPrp28 (residues 183–462) with hPrp28 (residues 402–689) beginning from where the structural data are available. Note that, yPrp28 structure is available only from residues 127–588 and hPrp28 structure from residues 352–806;
 - (3) Annotated structure of human Prp28 within the solved human Pre-B complex. Note that the corresponding BPA-replaced residues in yeast Prp28 is marked onto this human structure, according to the preceding sequence alignment. The PyMOL file of this analysis is also uploaded.
- (2) In Supplementary Tables 3 (which lists the Prp28-K136(BPA)-crosslinked proteins identified by mass spec.) and 6 (which lists Prp28-E326(BPA)-crosslinked proteins identified by mass spec.) the common names of the proteins should be supplied in addition to the systematic names, when they exist.

Done. Please see **Supplementary Tables S5 and S8.**

- (3) In line 69 is Prp28 meant rather than Prp8?

Yes, sorry for the typo.

- (4) The authors should attempt to explain the strong crosslink between Prp28-K41(BPA) and Brr2 at 2 mM ATP, using the human pre-B structure or structures from later in the splicing cycle. Figure 4 is inadequate in this regard since Brr2 is far from Prp28 in every panel. **BPA is essentially a zero Angstrom crosslinker so near-atomic structures should be used to rationalize the results. It is not clear what the word “trans-relocation” on line 87 means.**

We have expanded our original writing by given more details and the underlying reasoning, hoping to have clarified this issue as follows:

“Two pre-B structures are now available. In the yeast structure, Prp28 cannot be precisely positioned. Our crosslinking data, though, appears to fit well with both the yeast and human pre-B structures with respect to the locations of Prp8 and Snu114. Yet, in the human Pre-B structure, Prp28 appears to be distant from the location of Brr2. In both cases, Brr2 is observed or predicted, respectively, to undergo a rotation and trans-relocation in the pre-B-to-B transition (Fig. 4). We note that in the yeast Pre-B structure, one of the predicted locations of Prp28 is close to Brr2 and that the human Brr2 is also physically close to Prp28 upon the predicted dramatic trans-relocation. We therefore speculate that this trans-relocation may then places Brr2 in the vicinity of Prp28. As the spliceosome complexes are highly dynamic during splicing process, our biochemical approach might have captured a structurally dynamic, but so far undetected, intermediate state.” (Page 5, Lines 119–129)

- (5) There is either a labeling or interpretation problem with Supplementary Figure 5 (right). The authors claim that two double mutants, prp28-K136del10 with either brr2-G1708* or -L1971*, manifest a suppressive interaction at 30°C (line 94). However, the double mutant with G1708* clearly grows slower than either single mutant, and the double mutant with L1971* is dead!

We apologize for this glaring error. With no exception, all pairs of tested mutations resulted in either synthetic lethality or sickness. We now deleted the mis-statement from the relevant sentence from the main text.

- (6) I disagree that a synthetic lethal phenotype supports the interactions identified, which is stated on line 95 and implied on line 133. If either mutation A or mutation B ablates an interaction, then the combination of the two should have no greater effect.

We concur. The two relevant sentences now read as follows:

“All double mutants exhibited synthetic-sick or -lethal phenotypes (Fig. 2a and Supplementary Fig. 5 and Supplementary Table 6), suggesting that Prp28 and Brr2 are functionally interacting with each other.”.

“Genetic analysis shows that the *prp28-E326Δ3 npl3Δ* double mutant exhibits a synthetic-lethal phenotype (Supplementary Fig. 8d). Immunoprecipitation analysis revealed that p-Npl3 associates with *ACT1* transcript in an ATP-independent fashion throughout the course of the splicing reaction (Fig. 3a).”

- (7) The Figure legend for Supplementary Figure 6 that starts at line 358 does not agree with the legend printed on the figure. The legend on the figure correctly states that the Y axis shows the amount of U1 snRNP detected (not “departed”) and that 100% is set by the wild-type reaction at 0.05 mM (not “2 mM”). **In general, there should not be two legends for a single figure in a submitted manuscript.** This is the case for all the supplementary figures.

The word “departed” in Supplementary Figure S6 has been corrected into “**detected**”. We now delete the Supplementary Figure legends from the main text, so that they only appear under the Supplementary Figures.

- (8) Line 145: “this ATPase activity is lost when Prp28-E326del3 is combined with p-Npl3”. “Lost” should be replaced with “diminished” since the ATPase activity in lane 9 is greater than in lanes 4 or 12 of Figure 3d/e. Are these differences significant?

The revised sentence now reads: “Critically, this ATPase activity is **diminished** when Prp28-E326Δ3 is combined with p-Npl3”.

According to our analysis, the differences in line 9, which is greater than in lines 4 or 12, is significant, suggesting that p-Npl3 can still enhance Prp28-E326Δ3 ATPase activity, presumably because deletion of three amino acids did not fully abolish the interaction between p-Npl3 and the 3-amino-acid deleted Prp28.

- (9) The authors should define “NET-2” solution since there are different formulations on the web.

This is now done in the revised Methods (section **Co-immunoprecipitation assays**).

- (10) Line 492: “10x specific activity” is not useful if 1x is not defined.

This is now clearly specified in the revised Methods (section **Splicing extracts, radioactively labeled RNA, and splicing assays**) as follows:

“Actin precursor RNA substrates were synthesized *in vitro* as runoff transcripts using SP6 RNA polymerase and labeled with [α -³²P] UTP at 2 mCi/ml, which was defined as 1X specific activity for the *ACT1* pre-mRNA substrate. The 10X specific activity was defined as substrates labeled with [α -³²P] UTP at 20 mCi/ml.”

- (11) If mass spectrometry of BPA-crosslinked proteins is described in ref. 30, that should be stated. If not, it should be referenced or described.

In the revised manuscript, **Reference 18** (originally called Reference 30) describes, in a step-by-step manner, on how to perform a large-scale BPA-crosslinking experiment. However, it does not include the mass spectrometry procedure, which is now added into the revised Methods (under **Mass spectrometry analysis**), also in great details.

REVIEWER COMMENTS

Reviewer #1 (Remarks to the Author):

The authors have satisfactorily addressed all of my comments and questions. I find the revised manuscript quite sound and a very interesting read. I have no further comments or questions regarding this manuscript.

Reviewer #2 (Remarks to the Author):

I think the authors have done an excellent job in addressing the reviewer's concerns and have no further useful feedback to offer at this time. This is a very solid piece of work that helps understand how spliceosome assembly is regulated at a critical step and is a strong contribution to the field.

Reviewer #3 (Remarks to the Author):

The revised manuscript from Yeh et al. is substantially improved. However, there are still deficiencies in the presentation of the results that should be addressed before publication. These are enumerated below.

1) I cannot find any reference to Supplementary Figures 1b-1g in the main text. These figures should be explained in the text. Supplementary Figures 1c and 1d are not adequately described in the figure legend. The first sentence is not complete. What is the location of the amplicon within the PMA1 gene? What does the filled bar labeled "PMA1" represent? Why is the Prp28 signal over the intronless PMA1 gene higher than over the ACT1 3' splice site? What is the difference between panels c and d?

2) As far as I can tell, no data are shown supporting BPA-crosslinks at four residues in the RecA domains (see Figure S2 and Table S4, residues Q268, G319, D348, and E404). These four crosslinks should not be reported in the absence of supporting data.

3) It appears (in Figure 1c and 1d) that Prp28 containing K82 substituted with BPA crosslinks to Prp8 at 2 mM ATP. This fact should be noted in Figure 1e, or it should be explained why it is disregarded.

4) Line 123: Supplementary Figure 2c should be cited at the end of the sentence. In the PyMOL file that corresponds to this figure the chains are not correctly labeled. For example, some Sm subunits are labeled as Lsm's, and Prp28 might not be recognized as hDDX23. Also, I suggest that U1 snRNA and the pre-mRNA be added to Supplementary Figure 2c for clarity.

5) In discussing the Prp28 N-terminus/Brr2 crosslinks (lines 120-129), the authors might note that the N-terminus of hBrr2 closely approaches the N-terminus of hPrp28 in the pre-B complex structure shown in Supplementary Figure 2c. Therefore, the "trans-relocation" of Brr2, which occurs after the pre-B state, may not be necessary for this crosslink to form.

6) The phenotypes listed in the second column of Table S7 will not be meaningful to most readers in the absence of some explanation. At a minimum, the papers in which the phenotypes of these alleles were established should be cited. Also, the last column has an incomplete label since the strain also has a PRP28 deletion.

7) The Methods section could use additional editing to remove redundant or unnecessary text.

8) Table S1 lacks the source and citations for strains not made by the authors, such as YJU75.

9) Table S2 does not include all plasmids used in the study (e.g., PRP8 plasmids) and does not provide references for plasmids obtained from other sources.

10) Lines 406-409: References should be supplied for all antibodies obtained from other labs.

11) Line 421 (and elsewhere): 2 mCi/ml is not a specific activity, it is a radionuclide concentration. Specific activity should have moles as the denominator. For example, 3000 Ci/mmol (see lines 649/650).

12) The description of recombinant protein production starting on line 600 contains no volumes, but should throughout.

13) I think Figure 4 could be improved to more accurately reflect the relative positions of splicing factors at the indicated stages of splicing. In addition, I doubt Npl3 is binding to the 5' splice site in the B complex.

Reviewer #4 (Remarks to the Author):

Referee's report on "Activation of Prp28 ATPase by Phosphorylated Npl3 at a Critical Step of Spliceosome Remodeling" by Yeh et al.

First, I would like to apologize for the delay in reviewing part of the above manuscript. I was approached by the editors, who requested me to evaluate the UV crosslinking data in this manuscript. This I was happy to do. However, I had difficulties in accessing the mass-spectrometric data uploaded into the repository PRIDE, which led to a delay.

In the work described in this manuscript, Yeh et al. generated various Prp28 protein variants in which several amino-acid residues were replaced by p benzoyl-phenylalanine (BPA, a photoactivatable unnatural amino acid). Yeast extracts containing the variants were UV-crosslinked at 365 nm. Immunoblotting revealed the proteins Brr2, Prp8, Snu114, U1-C and Npl3 crosslinked predominantly to Prp28. Importantly, the observed crosslinking pattern was strictly dependent on the presence of pre mRNA and ATP and was not sensitive to treatment with RNase A.

The presence of protein Prp8 as the major crosslinked protein to the Prp28-K136BPA protein was confirmed by mass spectrometric (MS) analyses after in-gel digestion of the band/region where the most prominent western-blot signal of Prp28-K136BPA and Prp8 was observed.

Overall, I consider the crosslinking experiments described and the results obtained through immunoblotting and quantitative MS to be largely consistent.

Nonetheless, there are some points regarding the MS analyses in this work that I would like to suggest should be addressed/discussed by the authors:

1. My first point concerns the MS-based data obtained from the crosslinked sample containing Prp28-K136BP. The authors compared the abundance of proteins in UV-irradiated and non-irradiated samples. The data are summarized in Supplementary Table S5. I also re-evaluated these data from the dataset uploaded to the PRIDE repository. I have attached an Excel sheet, with the corresponding data evaluation, to this review ("LFQ_Prp28K136BPA_reviewer.xlsx). I compared label-free quantification (LFQ, as performed by the authors) with values from iBAQ. Indeed, Prp8 is the protein that is most abundantly crosslinked in the sample containing Prp28-K136BPA, supporting the assertion of a predominant crosslink between Prp28-K136BPA and Prp8. Also Brr2, Snu114 are enriched and thus suggest that Prp28-K136BPA also crosslinks to both these proteins. However, U1C was not

identified by MS. The authors might comment on this. Furthermore, Npl3 was also found to be much more abundant in one replicate of the two UV-irradiated sample of Prp28-K136BPA; this is obvious when one considers iBAQ values when compared with LFQ values (lane 33 of the attached Excel table "LFQ_Pr28K136BPA_reviewer"), "inf" (LFQ) and 3,8 (iBAQ). The LFQ values lead to "inf" (infinite values), which means that potential Prp28-K136BPA crosslinks are observed in UV-irradiated samples only. The authors should comment on why Npl3 is not listed in Supplementary Table S4 with Prp28K136BPA crosslinked and in Table S5b, where (respectively) the crosslinks of proteins to the various Prp28-aaBPA constructs and the top 10 most enriched proteins in the UV-irradiated Prp28K136BPA sample are listed. Furthermore, the authors cut out the band/region corresponding to the size of a crosslinked Prp28K136BPA-Prp8 moiety (clearly above MW 250 kDa, Figure 3d and Supplementary Figures S4b and c). The authors should explain why proteins of very different MW (Snu114, a crosslinked band migrating clearly below a MW of 250 kDa (Figure 3d) and Npl3, a crosslinked band (shown only for p-Npl3 in Supplementary Figure S8) migrating also clearly below 250 kDa are also found in this MW region on an SDS-polyacrylamide gel. I also wonder why the authors did not perform a similar MS analyses of the protein region containing the Prp28K136BPA-U1C crosslink (Figure 1d) and Prp28K136BPA-Sun114 (Figure 1d), both of which clearly migrate at a MW below that of Prp28-K136BPA-Prp8/Brr2.

2. Another point that should be raised is along the same lines as that made by reviewer #1, namely, the association of p-Npl3 with Prp28. This is an important issue, as it seems that exclusively p-Npl3 associates with Prp28, and, as the model outlined in Figure 4 suggests that, in the action of phosphorylated Npl3 (p-Npl3), this protein associates with U1 snRNP, yet it remains bound to the spliceosome after the Prp28-induced dissociation of U1 snRNP from the spliceosome. I agree with reviewer #1 that these data are not entirely compelling, and must add that the answers of the authors in their point-by-point reply are also not entirely satisfying.

The authors show in Supplementary Figure S8 that p-Npl3 crosslinks to Prp28-E326BPA by using a p-Npl3-specific antibody. The authors also demonstrated in *in vitro* co-affinity purifications that Prp28 only binds p-Npl3, and not non-phosphorylated Npl3. The manuscript would be significantly strengthened if the authors can show this unambiguously in the context of the spliceosome. Unfortunately, the relevant data, as presented, are only of indirect nature. At the end of my report I have suggested some experiments that the authors might consider (see point 2.6.).

2.1. Surprisingly, although stated in the main text of the manuscript (page 8), the MS data from a similar analysis of Prp28-E326BPA crosslinked to Npl3 were not found in the uploaded MS data in the PRIDE repository. The authors must provide these data.

2.2. A more critical issue is to determine the overall phosphorylation state of Npl3 in the spliceosome. Here the authors should perform phosphopeptide enrichment and MS analysis from a spliceosome sample (see also below). Otherwise Supplementary Table S9, in which the phosphorylation state, presumably of Npl3 co-expressed in the presence of the kinase Sky1, would not be very compelling (see below).

2.3. The authors provided in Supplement Table S9 a list with all serine and tyrosine residues of Npl3 between positions 306 and 411 that are phosphorylated and the corresponding localization probability according to MaxQuant (MQ) software. I also re-evaluated these data uploaded in PRIDE. Overall, the localization probability is correct (see Table "phospho_sites_npl3_reviewer"), with minor exceptions: A few serine and tyrosine residues are located in repetitive sequences. Here the localization probability is ambiguous (see Table "phospho_sites_npl3_ambiguous_reviewer" attached to my report). The authors might check this and adapt their table accordingly.

2.4. The localization probability is nonetheless very useful, however, and the degree of phosphorylation of every amino-acid residue is also (in fact, even more) important. Therefore the authors should also list quantitative values: to what degree (percentage) each amino acid is phosphorylated compared with the other phosphorylated residues (e.g. from a quantitative comparison of the extracted ion chromatograms of the various phosphorylated peptides over the total ion chromatogram).

2.5. The authors did not provide information on how exactly the MS of phosphopeptides from Npl3 was performed. The Materials and Methods section only contains a description of how the protein identification of the crosslinks of Prp28-K136BPA samples was performed. There, it is stated that the data were also searched against phosphorylated Y, T and S residues. However, this particular analysis probably cannot yield the phosphorylated residues listed in supplementary Table S9, because (i) without any enrichment of phosphorylated peptides prior to MS analyses barely any phosphopeptide can be identified from a complex mixture, (ii) Npl3 was identified in crosslinked sample (Prp28K136BPA) but with peptides not matching the RS-domain (see attached table "peptides_npl3_prp28k136bpa_reviewer") and (iii) as mentioned above, the important MS data regarding Prp28E326BPA are not available. I reckon that the phosphopeptide MS analysis presented (Supplementary Table S9) was done from the same batch of phosphorylated Npl3 that was used in the ATP hydrolysis assay or in the in vitro co-affinity studies. The authors must provide a clear description of which sample(s) was/were used and of exactly how the phosphopeptide MS analysis was done (N.B. Also in the data uploaded to PRIDE there is no description of the corresponding experiments).

2.6. In their western-blot analyses the authors show that presumably only p-Npl3 is crosslinked to Prp28-E326BPA. As mentioned above, Npl3 is also enriched in the crosslink of Prp28K136BPA, however, here no information of the phosphorylation status of Npl3 can be obtained by MS. Furthermore, MS data of the crosslinking between Prp28E326-(p-)Npl3 are not available. In my opinion, this study would benefit from additional experiments that, in addition to the Western blot analyses, provide additional proof for the interaction/crosslink of p-Npl3 to Prp28(E326BPA). For example: (i) The authors should perform MS analyses of enriched phosphopeptides derived from (crosslinked) spliceosomes containing Npl3 (see above, point 2.2.). (ii) I wonder whether the authors might perform an experiment using Prp28E326BPA-crosslinked spliceosomes and treat these with phosphatase to remove phosphorylation sites in Npl3 and then monitor the presence or absence of (p-)Npl3 with p-Npl3 and Npl3 antibodies. (iii) An experiment similar to (ii) but phosphatase treatment before crosslinking. (iv) Would it be possible to show that p-Npl3 is not present in the spliceosome in a Prp28-E326 Δ 3 mutant strain or, conversely, in an npl3-S411A mutant strain. (iv) I also wonder why the authors have not shown, in their in vitro co-affinity purification experiments with Prp28 and Npl3 and p-Npl3, that a Prp28-E326 Δ 3 protein does not bind to p-Npl3 or, conversely, that Npl3-S411A does not bind to Prp28(E326 Δ 3).

All in all, since the phosphorylation of p-Npl3 and its interaction with Prp28 are the major findings, the authors should strengthen the experimental basis of these findings, and of their conclusion as suggested in my points above.

REVIEWER #3:

- (1) I cannot find any reference to Supplementary Figures 1b-1g in the main text. These figures should be explained in the text. Supplementary Figures 1c and 1d are not adequately described in the figure legend. The first sentence is not complete. What is the location of the amplicon within the PMA1 gene? What does the filled bar labeled “PMA1” represent? Why is the Prp28 signal over the intronless PMA1 gene higher than over the ACT1 3’ splice site? What is the difference between panels c and d?

As suggested, references to Supplementary Figures 1b-g are now well placed in the main text (line 75 and elsewhere; all highlighted in red).

Supplementary Figures 1c and 1d represent two separate ChIP experiments for exactly the same purpose, i.e., probing Prp28’s association along the length of the *ACT1* transcript, but using two different sets of primers (A–D [for 1c] and 1–6 [for 1d]). And they yielded essentially the same conclusion. To avoid confusion and to eliminate redundancy, we decide to remove the “old” 1c panel and show only the “old” 1d panel (now, the “new” 1c). Another critical reason to do so is because the 1d approach offers finer resolution (6 sets of primers) than that of 1c (four sets of primers). In this way, any confusion of PMA1 is no longer an issue.

- (2) As far as I can tell, no data are shown supporting BPA-crosslinks at four residues in the RecA domains (see Figure S2 and Table S4, residues Q268, G319, D348, and E404). These four crosslinks should not be reported in the absence of supporting data.

As suggested, we now include the BPA-crosslinking results of Q268, G319, D348, and E404 in the Supplementary Figures 1g, which is also referenced in lines 95-96 in the main text (highlighted in red).

- (3) It appears (in Figure 1c and 1d) that Prp28 containing K82 substituted with BPA crosslinks to Prp8 at 2 mM ATP. This fact should be noted in Figure 1e, or it should be explained why it is disregarded.

Yes, indeed. This fact is now clearly added into the Figure 1e (i.e., the summary diagram) as a small K82-box under a Prp8 red circle (shown horizontally on the top), which corresponds to 2 mM ATP (shown vertically on the top).

- (4) Line 123: Supplementary Figure 2c should be cited at the end of the sentence. In the PyMOL file that corresponds to this figure the chains are not correctly labeled. For example, some Sm subunits are labeled as Lsm’s, and Prp28 might not be recognized as hDDX23. Also, I suggest that U1 snRNA and the pre-mRNA be added to Supplementary Figure 2c for clarity.

First, reference to Supplementary Figure 2c is now added to the end of the first sentence in Line 124 (highlighted in red in the main text), which deals with the description of human spliceosome structure.

Second, as suggested, we corrected the labels in the PyMol file, which yielded a new Supplementary Figure 2c. In this new figure, Sm, hDDX23 (hPrp28), U1 snRNA, and the AdML pre-mRNA are clearly displayed.

- (5) In discussing the Prp28 N-terminus/Brr2 crosslinks (lines 120-129), the authors might note that the N-terminus of hBrr2 closely approaches the N-terminus of hPrp28 in the pre-B complex structure shown in Supplementary Figure 2c. Therefore, the “trans-relocation” of Brr2, which occurs after the pre-B state, may not be necessary for this crosslink to form.

The N-terminus of human Prp28 is indeed close to human Brr2. However, this human Prp28 N-terminus is not present (or conserved) in the yPrp28 (discussed extensively in the main text). As a result, the amino-acid residues (K27, K41, K82, and K136) in the yeast Prp28 we found to crosslink to the yeast Brr2 are, by prediction, correspond to a region of human Prp28 that is distant from the human Brr2. For this reason, we proposed a “trans-relocation” scenario. We now include this argument in the main text, which reads as follows:

“Yet, in the human Pre-B structure, the bulk part of hPrp28 except its N-terminal region, which is not conserved in yPrp28 (see below), appears to be distant from the location of Brr2 (Supplementary Figure 2c). The amino-acid residues in yPrp28 (K27, K41, K82, and K136) that crosslink to Brr2 would then be predictably structurally distant from Brr2”. (Lines 122–126)

- (6) The phenotypes listed in the second column of Table S7 will not be meaningful to most readers in the absence of some explanation. At a minimum, the papers in which the phenotypes of these alleles were established should be cited. Also, the last column has an incomplete label since the strain also has a PRP28 deletion.

As suggested, we now provide a list of source papers describing the phenotypes of all these *prp8* alleles (shown at the bottom of Table S7). The label “Synthetic Lethality *YHC1-1 prp28Δ*” is now correctly stated in the top row of this Table, next to Source/Reference.

- (7) The Methods section could use additional editing to remove redundant or unnecessary text.

We have done so.

- (8) Table S1 lacks the source and citations for strains not made by the authors, such as YJU75.

As requested, we have added the source and citations accordingly under the column of Source/Reference. Cited references can be found at the bottom of the Table S1.

- (9) Table S2 does not include all plasmids used in the study (e.g., PRP8 plasmids) and does not provide references for plasmids obtained from other sources.

We have done exactly as requested in the revised Table S2, which now includes a huge section of *PRP8* plasmids.

- (10) Lines 406-409: References should be supplied for all antibodies obtained from other labs.

As requested, specific information about all antibodies are provided in the Methods section under the subtitle of “Antibodies and reagents”.

- (11) Line 421 (and elsewhere): 2 mCi/ml is not a specific activity, it is a radionuclide concentration. Specific activity should have moles as the denominator. For example, 3000 Ci/mmol (see lines 649/650).

This is now corrected in Methods. For example: “... transcript...labeled with [α - 32 P] UTP at 20 Ci/mmol, which was defined as 1X specific activity for the *ACT1* pre-mRNA substrate...”

- (12) The description of recombinant protein production starting on line 600 contains no volumes, but should throughout.

As suggested, we have added the starting volume for recombinant protein production (See Methods). For example: “We typically grew 6 liters of *E. coli* culture for purifying recombinant Prp28^{WT} or Prp28^{AAAD} protein.”

- (13) I think Figure 4 could be improved to more accurately reflect the relative positions of splicing factors at the indicated stages of splicing. In addition, I doubt Npl3 is binding to the 5' splice site in the B complex.

As suggested, we have redrawn Figure 4 in accordance to the displayed orientations of the yeast pre-B and B complexes (Bai *et al.*, *Science* [2018]). In this new drawing, as pointed by the reviewer, we have moved p-Npl3 away from the 5' splice site to avoid confusion.

REVIEWER #4:

Overall, I consider the crosslinking experiments described and the results obtained through immunoblotting and quantitative MS to be largely consistent (Thank you!). Nonetheless, there are some points regarding the MS analyses in this work that I would like to suggest should be addressed/discussed by the authors:

- (1) My first point concerns the MS-based data obtained from the crosslinked sample containing Prp28-K136BP... I also re-evaluated these data from the dataset uploaded to the PRIDE repository... I compared label-free quantification (LFQ, as performed by the authors) with values from iBAQ. Indeed, Prp8 is the protein that is most abundantly crosslinked in the sample containing Prp28-K136BPA, supporting the assertion of a predominant crosslink between Prp28-K136BPA and Prp8. Also Brr2, Snu114 are enriched and thus suggest that Prp28-K136BPA also crosslinks to both these proteins. However, U1C was not identified by MS. The authors might comment on this.

The key reason why U1C was not detected by mass-spec analysis is because the way we did the experiment. We focused only on the most abundant Prp28 (67 Kd)-crosslinked species. In the case of Prp28-K136BPA, the most dominant species turned out to be Prp28-Prp8 (~300 Kd), which run close to the top of the gel. We then excised this species in a small gel slice for mass-spec analysis. This gel slice is expected to be free from Prp28-U1C crosslinked product ($67 + 22 = 89$ Kd), which should run way below the cut-out ~300 Kd (= Prp28-Prp8) band.

So, how did we know Prp28 crosslinks with U1C? From our previous published papers, we know U1C is most likely a target of Prp28 (Chen *et al.* [2001] *MolCell*; Hage *et al.* [2009] *MCB*). Naturally, we probed for the Prp28-U1C crosslinked species using anti-U1C antibody. Nonetheless, that was not done by mass-spec analysis.

- (2) Furthermore, Npl3 was also found to be much more abundant in one replicate of the two UV-irradiated sample of Prp28-K136BPA...The authors should comment on why Npl3 is not listed in Supplementary Table S4 with Prp28K136BPA crosslinked and in Table S5b, where (respectively) the crosslinks of proteins to the various Prp28-aaBPA constructs and the top 10 most enriched proteins in the UV-irradiated Prp28K136BPA sample are listed.

We agree with the reviewer. Accordingly, we now include Npl3 in the updated Table S4. In addition, we also updated Table S5b, in which RNA-processing-related proteins (including Npl3) detected by mass spec are pooled together, for clarity sake.

To go one step further and just to be sure, we conducted new Prp28K136BPA UV crosslinking reactions and probed with anti-p-Npl3 antibody. The result indicates that K136BPA indeed directly interacts with p-Npl3 (**Supplementary Figure 4d**).

- (3) The authors should explain why proteins of very different MW (Snu114, a crosslinked band migrating clearly below a MW of 250 kDa (Figure 3d) and Npl3, a crosslinked band (shown only for p-Npl3 in Supplementary Figure S8) migrating also clearly below 250 kDa are also found in this MW region on an SDS–polyacrylamide gel. I also wonder why the authors did not perform a similar MS analyses of the protein region containing the Prp28K136BPA-U1C crosslink (Figure 1d) and Prp28K136BPA-Sun114 (Figure 1d), both of which clearly migrate at a MW below that of Prp28-K136BPA-Prp8/Brr2.

As to why the crosslinked species smaller than 250 Kd would end up in the cut-out gel slice, we surmise that it could be because the crosslinked species do not necessarily adopt an anticipated linear denatured form. In other words, the chemical crosslinks may render them a different shape (e.g., X- or Y-shape), thus getting retarded more in the gel system.

As explained above, we focused only on the most abundant (and clean) Prp28 (67 Kd)-crosslinked species for mass-spec analysis. For other potential interacting proteins, such as Brr2, Snu114, and U1C, we established them by intelligent guesses based on prior studies and literature using corresponding antibodies. This point was clearly stated in the main text as follows: “*On the basis of a combination of crosslinked species’ molecular sizes, Prp8’s*

location in published structures^{20,21}, and Prp28's known genetic interactions^{6,8}, we systematically interrogated other crosslinked proteins using a panel of antibodies. This effort identified two additional U5-snRNP proteins, Brr2 and Snu114, as well as UIC (Fig. 1d and Supplementary Fig. 3)" (Lines 104–108).

Authors' note: Hereafter, Reviewer 4 raised a series of interconnected comments/questions centering on the phosphorylation state of Npl3 within the splicing complexes. **To more clearly and logically respond to Reviewer 4, we decide to re-order the comments/questions, beginning from those we have taken care of and then address several comments/questions altogether.**

- (i) Surprisingly, although stated in the main text of the manuscript (page 8), the MS data from a similar analysis of Prp28-E326BPA crosslinked to Npl3 were not found in the uploaded MS data in the PRIDE repository. The authors must provide these data.

The original Prp28-E326PBA crosslinking experiment was done quite some years ago by a commercial entity, who at the time provided us with only processed data. Unfortunately, their practice was to keep the raw data for only three months on their server. For this reason and as requested by the Reviewer, we decided to repeat the whole Prp28-E326PBA crosslinking and mass-spec analysis. These MS data has been uploaded to PRIDE repository and summarized in **Table S8**. The silver-staining gel image and validation of p-Npl3 by Western analysis are shown in a new **Supplementary Figure 8**.

- (ii) The authors provided in Supplement Table S9 a list with all serine and tyrosine residues of Npl3 between positions 306 and 411 that are phosphorylated and the corresponding localization probability according to MaxQuant (MQ) software. *I also re-evaluated these data uploaded in PRIDE. Overall, the localization probability is correct* (see Table "phospho_sites_npl3_reviewer"), with minor exceptions: A few serine and tyrosine residues are located in repetitive sequences. Here the localization probability is ambiguous (see Table "phospho_sites_npl3_ambiguous_reviewer" attached to my report). The authors might check this and adapt their table accordingly.

The localization probability is nonetheless very useful, however, and *the degree of phosphorylation of every amino-acid residue* is also (in fact, even more) important. Therefore the authors should also list quantitative values: to what degree (percentage) each amino acid is phosphorylated compared with the other phosphorylated residues (e.g. from a quantitative comparison of the extracted ion chromatograms of the various phosphorylated peptides over the total ion chromatogram).

We sincerely thank the Reviewer for providing an exemplary Table format for better presentation. In accordance, we have updated Supplementary **Table S9a**. In addition, we include a new **Table S9b** summarizing the LC-MS Quantification of Site-Specific Phosphorylation Degree of p-Npl3), the data of which were calculated by using Xcalibur software and each calculation was displayed in **Supplementary Figure 11**.

- (iii) The authors did not provide information on how exactly the MS of phosphopeptides from Npl3 was performed. The Materials and Methods section only contains a description of how the protein identification of the crosslinks of Prp28-K136BPA samples was performed. There, it is stated that the data were also searched against phosphorylated Y, T and S residues. However, this particular analysis probably cannot yield the phosphorylated residues listed in supplementary Table S9, because (i) without any enrichment of phosphorylated peptides prior to MS analyses barely any phosphopeptide can be identified from a complex mixture, (ii) Npl3 was identified in crosslinked sample (Prp28K136BPA) but with peptides not matching the RS-domain (see attached table “peptides_npl3_prp28k136bpa_reviewer”) and (iii) as mentioned above, the important MS data regarding Prp28E326BPA are not available. I reckon that the phosphopeptide MS analysis presented (Supplementary Table S9) was done from the same batch of phosphorylated Npl3 that was used in the ATP hydrolysis assay or in the in vitro co-affinity studies. The authors must provide a clear description of which sample(s) was/were used and of exactly how the phosphopeptide MS analysis was done (N.B. Also in the data uploaded to PRIDE there is no description of the corresponding experiments).

In response to Reviewer’s comments, we have added more detailed descriptions in Methods (see below) and uploaded the corresponding experimental data to PRIDE.

“Purified recombinant protein (1.2 µg) of p-Npl3 was resolved by SDS-PAGE and Coomassie-Blue stained, then the single band was retrieved and analyzed for phosphorylation position by MS analysis. (Supplementary Figure 10d). The phosphopeptide MS analysis presented was done from the same batch of phosphorylated Npl3 that was used in the ATP hydrolysis assay and in the in vitro co-affinity studies.”

“In addition, variable modification of phospho (STY) for the phosphopeptide identification was analyzed and the localization probabilities were filtered larger than 0.95”.

We shall address the following comments/questions all together, because they are interconnected and center on the phosphorylation state of Npl3 within the splicing complexes.

- (a) Another point that should be raised is along the same lines as that made by reviewer #1, namely, the association of p-Npl3 with Prp28. This is an important issue, as it seems that exclusively p-Npl3 associates with Prp28, and, as the model outlined in Figure 4 suggests that, in the action of phosphorylated Npl3 (p-Npl3), this protein associates with U1 snRNP, yet it remains bound to the spliceosome after the Prp28-induced dissociation of U1 snRNP from the spliceosome.
- (b) The authors show in Supplementary Figure S8 that p-Npl3 crosslinks to Prp28-E326BPA by using a p-Npl3-specific antibody. The authors also demonstrated in in vitro co-affinity purifications that Prp28 only binds p-Npl3, and not non-phosphorylated Npl3. The manuscript would be significantly strengthened if the authors can show this unambiguously in the context of the spliceosome. Unfortunately, the relevant data, as presented, are only of indirect nature. At the end of my report I have suggested some experiments that the authors might consider (see point 2.6.).

- (c) A more critical issue is to determine the overall phosphorylation state of Npl3 in the spliceosome. Here the authors should perform phosphopeptide enrichment and MS analysis from a spliceosome sample (see also below). Otherwise Supplementary Table S9, in which the phosphorylation state, presumably of Npl3 co-expressed in the presence of the kinase Sky1, would not be very compelling (see below).
- (d) In their western-blot analyses the authors show that presumably only p-Npl3 is crosslinked to Prp28-E326BPA. As mentioned above, Npl3 is also enriched in the crosslink of Prp28K136BPA, however, here no information of the phosphorylation status of Npl3 can be obtained by MS. Furthermore, MS data of the crosslinking between Prp28E326-(p-)Npl3 are not available. In my opinion, this study would benefit from additional experiments that, in addition to the Western blot analyses, provide additional proof for the interaction/crosslink of p-Npl3 to Prp28(E326BPA). For example: (i) The authors should perform MS analyses of enriched phosphopeptides derived from (crosslinked) spliceosomes containing Npl3 (see above, point 2.2.). (ii) I wonder whether the authors might perform an experiment using Prp28E326BPA-crosslinked spliceosomes and treat these with phosphatase to remove phosphorylation sites in Npl3 and then monitor the presence or absence of (p-)Npl3 with p-Npl3 and Npl3 antibodies. (iii) An experiment similar to (ii) but phosphatase treatment before crosslinking. (iv) Would it be possible to show that p-Npl3 is not present in the spliceosome in a Prp28-E326 Δ 3 mutant strain or, conversely, in an npl3-S411A mutant strain. (iv) I also wonder why the authors have not shown, in their in vitro co-affinity purification experiments with Prp28 and Npl3 and p-Npl3, that a Prp28-E326 Δ 3 protein does not bind to p-Npl3 or, conversely, that Npl3-S411A does not bind to Prp28(E326 Δ 3).

All in all, since the phosphorylation of p-Npl3 and its interaction with Prp28 are the major findings, the authors should strengthen the experimental basis of these findings, and of their conclusion as suggested in my points above.

We deeply acknowledge Reviewer's critiques. After receiving the critiques, we have tried hard on the related experiments but failed to obtain high-quality data. To us (i.e., no mass-spec technical experts), this may not be totally unexpected, because the said experiments demand repeated and prolonged tries to obtain publishable data.

In retrospect, the mass-spec data was just a starting point for us to march onto this extensive (if not epic) landscape that covers numerous technically challenging biochemical and genetic experiments. As a result, we decided, at this point, to tone down the claim and discuss the limitation as suggested by the Editor as follows:

Immediately after the sentence "We therefore suggest that p-Npl3 is a cofactor that stimulates Prp28's ATPase activity to remove U1 snRNP from the pre-B complex" (Line 194–195), we now add a "tone-down-the-claim" sentence to acknowledge the limitation of our interpretation.

"However, the precise nature, such as the exact phosphorylation site(s), of such p-Npl3 within the splicing complexes remains to be determined." (Line 195-196)

REVIEWERS' COMMENTS

Reviewer #3 (Remarks to the Author):

The authors have adequately responded to my prior comments. I have just two final comments:

1) Re: my prior comment #1, there still appears to be no citations to Supplementary Figures 1c, 1d, 1e, and 1f in the main text.

2) I disagree with the authors' response to my prior comment #5. Based on the Nagai lab human pre-B spliceosome structure (6QX9) the most N-terminal residue of hBrr2 modeled (P36) is within 20 Å of hPrp28 residue R270. Alignment of the N-termini of yPrp28 and hPrp28 aligns hPrp28 R270 with yPrp28 N48 (see Teigelkamp et al. 1997; PMC1369570). The N-terminal yPrp28 crosslinks to yBrr2 observed in this study include residues in this region (K27, K41, K82, K136). Therefore, the N-terminus of yPrp28 is not "predictably structurally distant" from the N-terminus of yBrr2 in the yeast pre-B spliceosome, as stated by the authors, especially since an additional 35 N-terminal residues of hBrr2 are not modeled in the structure. Thus, yBrr2 need not be "transrelocated" for these crosslinks to form.

Reviewer #4 (Remarks to the Author):

Referee's report on the revised manuscript "Activation of Prp28 ATPase by Phosphorylated Npl3 at a Critical Step of Spliceosome Remodeling" by Yeh et al.

I appreciate the authors' reply on points 1 – 3 that I made in my review of their original manuscript. I further appreciate the authors' comment on my related comments/questions and their provision of supplementary Figures and Tables along with more detailed descriptions of the phosphopeptide analyses.

I am aware that the additional points regarding the precise phosphorylation state of Npl3 in the spliceosomal context would require extensive MS analyses. Additionally, I suggested experiments that might use biochemical methods (e.g. western blot) in order to find out the phosphorylation state, in particular in crosslinked Prp28 species. The authors were unable to address these points, presumably as the necessary experiments would have required much more time owing to their very sophisticated nature. The authors have therefore included a statement that the exact phosphorylation status of Npl3 in the spliceosomal context needs to be addressed.

All in all, it is my opinion that the authors have now addressed my concerns in a satisfactory manner, and I am happy to state that the revised version of the manuscript by Yeh et al. should be published in Nature Communications.

REVIEWER #3:

- (1) Re: my prior comment #1, there still appears to be no citations to Supplementary Figures 1c, 1d, 1e, and 1f in the main text.

We have now corrected this issue. These citations are located as follows: Supplementary Fig. 1c (Line 87); 1d (Line 100); 1e and 1f (Line 113).

- (2) I disagree with the authors' response to my prior comment #5. Based on the Nagai lab human pre-B spliceosome structure (6QX9) the most N-terminal residue of hBrr2 modeled (P36) is within 20 Å of hPrp28 residue R270. Alignment of the N-termini of yPrp28 and hPrp28 aligns hPrp28 R270 with yPrp28 N48 (see Teigelkamp *et al.* 1997; PMC1369570). The N-terminal yPrp28 crosslinks to yBrr2 observed in this study include residues in this region (K27, K41, K82, K136). Therefore, the N-terminus of yPrp28 is not “predictably structurally distant” from the N-terminus of yBrr2 in the yeast pre-B spliceosome, as stated by the authors, especially since an additional 35 N-terminal residues of hBrr2 are not modeled in the structure. Thus, yBrr2 need not be “transrelocated” for these crosslinks to form.

We agree with reviewer's comment. As a result, we decided to tone down our writing as follows:

“In the human Pre-B structure, the main body of hPrp28 appears to be distant from the location of Brr2 (Supplementary Fig. 2c and Supplementary Software 1). Yet, the N-terminal domain of hPrp28, which is not conserved in yPrp28 (see below), threads through Prp8 and Snu114 moieties to reach Brr2 (Supplementary Fig. 2c). In both human and yeast cases, Brr2 is observed²¹ or predicted²⁵, respectively, to undergo a rotation and trans-relocation in the pre-B-to-B transition (Fig. 4). In this light, we note that in the published yeast Pre-B structure, one of the speculated locations of Prp28 is close to Brr2 (Ref. ²¹), while the other location is not; whereas the human Brr2, upon the predicted dramatic trans-relocation²⁵, would also be physically close to the Prp28 main body. It is therefore tempting to speculate that this trans-relocation may then place Brr2 in the vicinity of Prp28 main body that is made up of the two RecA-like (i.e., the enzymatic) domains. At the moment, we cannot rule out that the crosslinking between Prp28 and Brr2 can occur without translocation, because the contact can be through the N-terminal domain of Prp28. However, as the spliceosome complexes are highly dynamic during splicing process, our biochemical approach might have captured a structurally dynamic, but so far undetected, intermediate state.” (Lines 132-146).

REVIEWER #4:

I appreciate the authors' reply on points 1–3 that I made in my review of their original manuscript. I further appreciate the authors' comment on my related comments/questions and their provision of supplementary Figures and Tables along with more detailed descriptions of the

phosphopeptide analyses. I am aware that the additional points regarding the precise phosphorylation state of Npl3 in the spliceosomal context would require extensive MS analyses. Additionally, I suggested experiments that might use biochemical methods (e.g. western blot) in order to find out the phosphorylation state, in particular in crosslinked Prp28 species. The authors were unable to address these points, presumably as the necessary experiments would have required much more time owing to their very sophisticated nature. The authors have therefore included a statement that the exact phosphorylation status of Npl3 in the spliceosomal context needs to be addressed. All in all, it is my opinion that the authors have now addressed my concerns in a satisfactory manner, and I am happy to state that the revised version of the manuscript by Yeh et al. should be published in Nature Communications.

We thank very much for Reviewer 4's valuable critiques that help to enhance our manuscript's scientific rigor.